# Estimation of degree of sea ice ridging in the Bay of Bothnia based on geolocated photon heights from ICESat-2

Renée Mie Fredensborg Hansen[1,2], Eero Rinne[1], Sinéad Louise Farrell[3], and Henriette Skourup[2]

[1]Finnish Meteorological Institute, Marine Research, Erik Palménin aukio 1, 00560 Helsinki, Finland
[2]DTU Space, Geodesy and Earth Observation, Elektrovej Building 328, 2800 Kgs. Lyngby, Denmark
[3]University of Maryland, Geographical Sciences, 2181 LeFrak hall, College Park, MD20740, United States

**Correspondence:** Renée Mie Fredensborg Hansen (renee.fredensborg@fmi.fi)

**Abstract.** We present a comparison of Ice, Cloud and land Elevation Satellite-2 (ICESat-2) geolocated photon heights and operational ice charts from the Finnish Ice Service in the Bay of Bothnia in spring 2019. We show that ICESat-2 (IS2) retrievals from ice areas with different ridging characteristics, more precisely degree of ridging (DIR), differ significantly. DIR is a particularly useful parameter for ice navigators, as it provides information on how difficult it is to navigate through an area based on e.g. sail heights and distribution of sea ice ridges. DIR estimates are included in ice charts of the Baltic Sea, and are based primarily on in situ observations from an active icebreaker fleet. We show that DIR may potentially be estimated from IS2. We also present a comparison of IS2 measurements and Sentinel-1 synthetic aperture radar frames, discussing several individual cases of IS2 photon elevation behaviour over Baltic sea ice. We suggest that IS2 data can be of benefit to international ice services, especially if a time critical photon height product were to be made available. Furthermore, we show that the difference between highest and mean photon elevations (elevation anomalies) of IS2 correspond to expected ridge sail heights in our study area. Our study is one of the first steps in creating sea ice applications beyond the traditional goal of freeboard and thickness retrieval for IS2.

## 1 Introduction

Rapid changes in the sea ice conditions cause challenges to ship navigation (Duncan et al., 2018). Thus, the ability to provide users with reliable and timely information on the ice conditions is time-critical and of high priority (Gegiuc et al., 2018). Some of the most important sea ice parameters for ice navigation are the ice extent, stage of development, concentration, thickness, and the amount and location of ridged ice. The amount and location of ridged ice is important, since navigation through heavily ridged sea ice is difficult and potentially dangerous (e.g., Kovacs et al., 1973; Gegiuc et al., 2018; Goerlandt et al., 2017; Ronkainen et al., 2018).

The Baltic Sea extends from 54 to 66° N with a total area of 422 000 $km^2$ (Ronkainen et al., 2018). The seasonal ice cover usually appears in early November and persists until mid-May with the largest extent between January and March (Goerlandt et al., 2017). Wintertime shipping through the ice covered sea into Northern harbours requires timely and accurate ice information provided by the Finnish and Swedish ice services. This information is provided in the form of daily operational ice charts. Generally, the sea ice in the Baltic Sea is divided into fast and drift ice. Fast ice occurs in coastal regions and archipelagos, and

grows thermodynamically as it is attached to the coast where it remains stationary (Ronkainen et al., 2018). Wind and currents drive the drift ice by moving the ice floes. Divergent motion forms cracks and leads in the ice cover, and convergent motion results in formation of rafted ice and ice ridges (Gegiuc et al., 2018).

Daily ice charts of the Baltic Sea ice are prepared by the Finnish Ice Service (FIS) analysts and provide a source of information on the ice conditions. The charts partition the ice cover into polygons to which ice types and other properties are assigned. Parameters assigned to each polygon are ice concentration, average level-ice thickness, maximum and minimum level thickness and the degree of ice ridging (DIR). Satellite synthetic aperture radar (SAR) imagery are the main data source for ice charts, but DIR is based mostly on visual icebreaker observations. This is because DIR is designed to be a representative description of the navigational difficulties from the point of view of the navigator (Ronkainen et al., 2018).

The National Aeronautics and Space Administration (NASA)'s Ice, Cloud and land Elevation Satellite-2 (ICESat-2) was launched on 15 September 2018 and builds upon the heritage of the Ice, Cloud and land Elevation Satellite (ICESat) mission (Neumann et al., 2019b; Abdalati et al., 2010). One of the primary mission objectives for ICESat-2 (IS2) is to estimate the thickness of sea ice and monitor any changes therein (Markus et al., 2017). The main payload of IS2 is the Advanced Topographic Laser Altimeter System (ATLAS), a photon counting laser system operated at 532 nm with a pulse-repetition frequency of 10 kHz (Kwok et al., 2019b). The ATLAS instrument employs a multi-beam configuration consisting of three pairs of beams (strong and weak beams) separated across-track by approximately 3 km and a pair spacing of 90 m (Brunt et al., 2019). The individual footprints of ∼17 m are separated by ∼0.7 m (Kwok et al., 2019a). The novel beam-pair photon-counting approach overcomes the limitations of its predecessor ICESat, as it allows for the determination of local across-track variations, e.g. sea surface height measurements in open, often narrow, leads required for sea ice freeboard and ice thickness retrievals (Markus et al., 2017). The dense surface coverage also allows for pressure ridge detection based on e.g. studies that used the photon elevations from Multiple Altimeter Beam Experimental Lidar (MABEL), an airborne simulator used to test the instrument theory and strategy of IS2 (Farrell et al., 2015). Furthermore, the first study using IS2 to estimate surface topography and ridges in the Arctic has recently been published (Farrell et al., 2020). IS2 is a novel instrument and we expect that there is a vast amount of information on sea ice in the data products not currently used due to lack of methodology. Developing such novel methodology is the key driver for this study.

This paper presents a feasibility study demonstrating the use of IS2 data (granules) to estimate sea ice ridging information relevant for ice navigation. Our study is based on four IS2 passes from early 2019 in the Baltic Sea. We compare IS2's Global Geolocated Photon Data (ATL03) product to ice charts from the FIS. We have chosen the Baltic Sea as our test area, since this is, to our knowledge, the only area covered by a dense time series of ice charts where ice ridging estimates are based on frequent reports from an active ice breaker fleet (WMO, 2010). We discuss the potential of IS2 to complement satellite SAR imagery which is widely used by ice services, and the potential benefits of a time-critical IS2 product to international ice services.

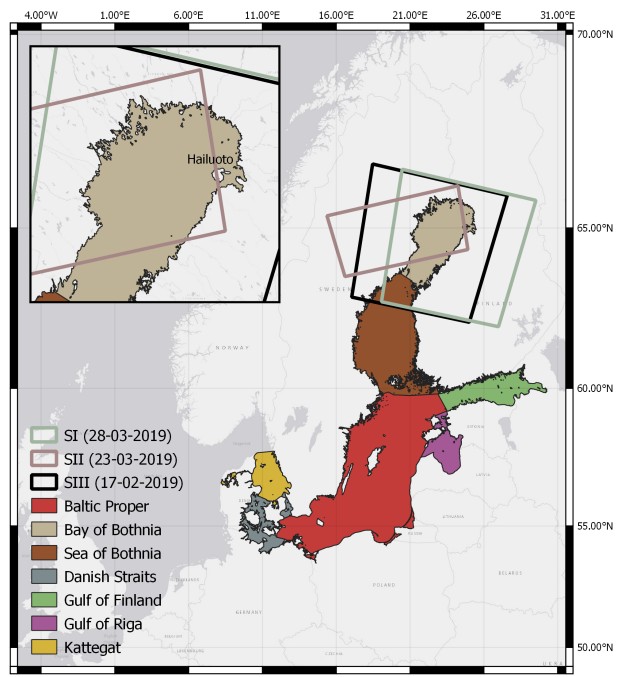

**Figure 1.** The Baltic Sea and its sub-basins as defined by the Baltic Marine Environment Protection Commission – Helsinki Commission (HELCOM) available at: http://maps.helcom.fi/website/mapservice/ (Accessed 21 October 2020). A highlight of our study area, the Bay of Bothnia, is inset top left. The synthetic aperture radar (SAR) frames utilised for cases, described in more detail in Table 1, are outlined as well by using the kml-files provided in the SAR product. Environmental Systems Research Institute (ESRI) World Base Light Grey is used for background map, available at: https://felix.rohrba.ch/en/download/2696/ (Accessed 21 October 2020).

## 2 Data and methods

Our study area (Fig. 1) is the northernmost basin of the Baltic Sea, the Bay of Bothnia, north of the sound of Quark at 63.5° N, during the ice season 2018–19. The ice winter of 2018–19 was mild with a maximum extent of the ice cover in the Baltic of 88 000 $\text{km}^2$ reached on 27 January. FIS classifies the winters as mild in the Baltic if the ice extent is below 115 000 $\text{km}^2$ (Ronkainen et al., 2018; BIM, 2019). The ship traffic was dense, where according to the Baltic Icebreaking Management (BIM), 1428 vessels were assisted during the 2018–19 ice season (27 December – 5 May) in the Bay of Bothnia (BIM, 2019). Even though the winter was mild, icebreakers frequently reported heavily ridged ice areas and rubble fields in February and March.

### 2.1 Ice charts and degree of ice ridging

FIS produces daily ice charts during the ice season. The ice charts are provided as both vector charts and gridded products that contain information on ice concentration, average thickness, minimum and maximum thickness, sea surface temperature

and a numeral description of deformation known as DIR (Gegiuc et al., 2018). DIR classifies ice into six categories (from 0 to 5) denoting level ice (0), rafted ice (1), slightly ridged ice (2), ridged ice (3), heavily ridged ice (4), and brash barrier
(5) (Ronkainen et al., 2018). The ice charts are based on satellite SAR, visible and infrared satellite imagery, sea ice models and in situ measurements including observations from ice breakers and coastal observations by volunteer ice observers (Berglund and Eriksson, 2015). Weekly ice charts including DIR are available in digital format since 1980. At the time of writing they are available only per request from the FIS. However, sea ice concentration and ice thickness from the FIS charts has been publicly available via the Copernicus Marine Service since July 2018 (https://resources.marine.copernicus.eu/?option=com_
csw&view=details&product_id=SEAICE_BAL_SEAICE_L4_NRT_OBSERVATIONS_011_004).

The quality and uncertainty of the ice chart depends partly on the quality of the SAR imagery, but also on the experience of the ice analyst, which both can result in inconsistencies in the final ice chart. Assigning a DIR value to each ice chart polygon is a complex process that requires a profound understanding of the current ice season and ice development (Gegiuc et al., 2018). Most importantly, routine reports from the Finnish and Swedish ice breaker fleets are used to estimate ridging. It should
be noted that assigning one numeral to a large area of sea ice is necessarily a simplification – in reality the ice in the area is always a mixture of several ice types. We emphasise that the FIS ice chart carries more information than just that from satellite SAR imagery. Because the high number of vessels in the Baltic at all times, FIS ice charts utilise significantly more in situ knowledge than Arctic ice charts. During the ice winter 2018–19, FIS received 1628 ice reports from ice breakers (BIM, 2019). For this reason they are used here as the reference data set.

DIR and satellite data have been compared to each other by Gegiuc et al. (2018). To build an automatic method to derive DIR from SAR imagery, they applied a random forest algorithm to dual-polarised (HH/HV) RADARSAT-2 SAR, where H denotes horizontal and V denotes vertical polarisation, and used the FIS ice charts as reference data. The results were promising when a significant amount of ridging had occurred, allowing for the ridging to strongly contribute to the texture of the SAR images. To the best of our knowledge, this is the only study that utilises satellite measurements to estimate DIR to supplement ice charting.
Analogously, we investigate the possibility to estimate DIR from IS2.

## 2.2 The Ice, Cloud and Land Elevation Satellite 2 (ICESat-2)

IS2's main payload, the laser altimeter ATLAS employs a multi-beam configuration consisting of six beams (3 beam pairs with a strong and weak beam). The strong beams are ∼4 times the pulse energy of the weak (Kwok et al., 2019a). Left (of individual beam pairs imaginary centerline) beam is denoted GTL, and right beams by GTR. Number of specific beam pair (dependent
on the orientation of the spacecraft) ranges from 1 to 3. The IS2 reference ground track (RGT) is an imaginary centerline of the ground track pattern of the multi-beam configuration, and it takes 91 days to sample all 1387 unique RGTs, fulfilling one full cycle (Brunt et al., 2019). One orbit track is divided into 14 granules (latitude-dependent regions) to limit data size to a maximum of 6 GB, such that the Bay of Bothnia appears in granule region 03 on ascending tracks (59.5–80°N) and granule region 05 on descending (80–59.5°N) (Neumann et al., 2020).

### 2.2.1 Global Geolocated Photons (ATL03) data product

In our study, we use the Global Geolocated Photons Level 2A Data Product (ATL03) from IS2 (Neumann et al., 2019a). ATL03 is produced by combining the laser pointing vectors, the position of the IS2 observatory and the individual photon times of flight from ATL02 (Science Unit Converted Telemetry Level 1B Data Product) (Neumann et al., 2019b). ATL03 includes the longitude (*lon_ph*), latitude (*lat_ph*) and World Geodetic System 1984 (WGS-84) ellipsoidal heights of the photons (*h_ph*) alongside a coarse discrimination of what is likely signal and what is background events (*signal_conf_photon*); a surface classification to identify land, ocean, land ice, sea ice and inland water (with surfaces overlapping by 20 km); geophysical corrections to be applied (Earth Gravitational Model 2008 (EGM2008) geoid (*geoid*), MOG2D dynamic atmosphere correction/inverted barometer as calculated by Archiving, Validation and Interpretation of Satellite Oceanographic data (AVISO) (*dac*) and ocean tide given by the GOT4.8 model (*ocean_tide*)) and other parameters useful for higher-level products (Neumann et al., 2019b).

The coarse discrimination of signal and background photons is based on generated along-track histograms. The identification of signal photon events is based on the location of regions where the photon event rate is significantly larger than the background photon event rate. All photons in a given bin are either classified as signal or background events. The planned data latency of ATL03 is 21 days, where latency is defined as the approximate time from data acquisition to data products reach the end users in a suitable format (Brown et al., 2016).

We found four overpasses with clear sky conditions in our study region where IS2 measured ice areas that were marked as ridged (DIR2–DIR4) in the FIS ice charts, i.e. on 1 February, 17 February, 23 March and 27 March 2019. Data from these dates are used in our study. The different DIR areas measured by IS2 as well as satellite ground tracks are shown in Sect. 3.1. DIR2 was sampled by IS2 on 1 February and 17 February, DIR3 on 23 March and 27 March, and DIR4 solely on 27 March.

### 2.3 Sentinel-1 (S1) C-band synthetic aperture radar (SAR) imagery

ESA launched the first satellite in the Copernicus Sentinel program (Sentinel-1A; S1A) in 2014, later joined by its twin satellite Sentinel-1B (S1B) in 2016. Flying in a twin-constellation tandem phase, they provide 6-day coverage of the Earth (individually 12-day repeat orbit), where they capture the surface using SAR imagery at C-band (5.4 GHz) (Kwok et al., 2019b). For this study, we utilise the Extra-Wide (EW) swath mode at high resolution (HR) of 50 × 50 m and swath width of 410 km. HR has a pixel spacing of 25 × 25 m in EW mode (Stasolla and Neyt, 2018). The S1A/B data were processed by ESA and archived at the Alaska Satellite Facility. In total, three SAR frames have been retrieved to discuss selected cases (Sect. 3.2), see overview provided in Table 3. All retrieved SAR frames (SI-SIII) were acquired in HV/HH polarisation.

### 2.4 Estimation of degree of ice ridging using ICESat-2 (IS2) photons

Our hypothesis is that the areas of heavier ridging (high DIR) should be distinguishable within the ATL03 photon elevations since the heavier the ridging, the higher the ridge sail heights and thus a larger amount of the measured elevations should be higher than the elevation of level ice, when compared to areas with less ridging. This assumption is only based on the amplitude of the ridging in the areas and not on spatial parameters, e.g. ridge density. Thus, we investigate the distributions

**Table 1.** Sentinel-1 (S1) frames utilised in this study with ID number for comparison with cases and dates for acquisition for both S1 and ICESat-2 (IS2)., see Table 3. Dates are provided in the UTC time frame.

| ID | Platform | Date (S1) | Date (IS2) | $\Delta t$ |
|------|----------|----------------|----------------|----------|
| SI | S1B | 28/03 04:57:15 | 27/03 18:22:53 | 10h35min |
| SII | S1B | 23/03 16:05:32 | 23/03 18:31:14 | 2h26min |
| SIII | S1B | 17/02 15:49:12 | 17/02 07:44:19 | 8h5min |

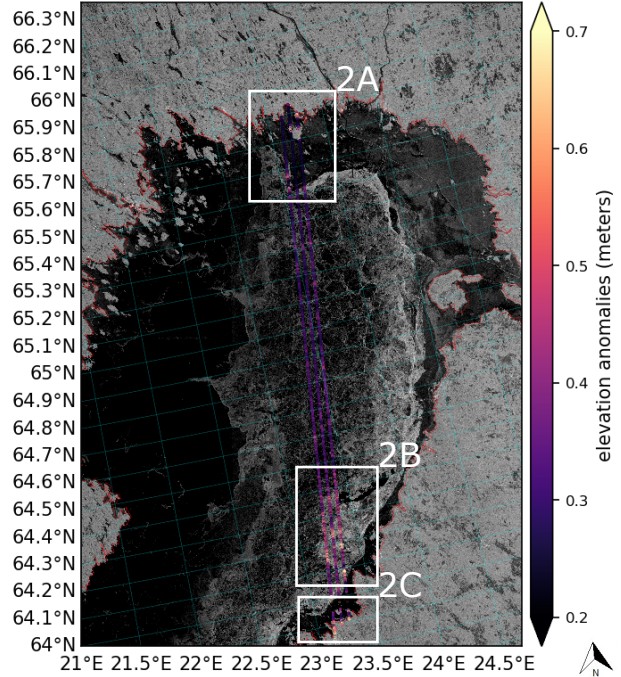

**Figure 2.** Elevation anomalies calculated using IS2 photons acquired 27 March 2019, overlaying S1 C-band (HV polarisation) SAR image acquired ~0.45d (10 hrs 35 min) after IS2. Note, the higher elevation anomalies acquired over more ridged ice (strong backscatter in white in SAR image). Areas mentioned in text are denoted 2A-2C.

of the photon elevations over a segment of $N = 150$ photons. This corresponds to segments of approximately 17 m in length (based on measurements of one strong beam (GT3L) from 17 February 2019, but exact segment length varies with the amount of geolocated photons reflected from the surface). We filter the ATL03 photons by selecting photons of only high signal confidence and apply the geophysical corrections. The geophysical correction is part of the pre-processing filtering. We also discarded all measurements that deviated from the geoid elevation by more than 3 m. On average, with the described pre-processing steps above (high signal confidence, +/- 3 metres from geoid elevation and applying a bounding box covering only the Bay of Bothnia), 19.86% (ranging from 8.32–38.65%) of the ATL03 photons were discarded. The largest amount of

photons were discarded on 1 February and 23 March 2019, where between 14.81–38.65% were removed in the pre-processing steps. Fewer were discarded on 17 February and 27 March ranging between 8.32–17.44%. We emphasise, that for these four days, most of the photons discarded were due to the +/- 3 metres requirement. Only few beams (3 strong beams on 23 March 2019) had photons discarded by the high signal confidence flag (0.01-0.02%, 32–54 photons). We further emphasise, that before actual pre-processing, all available tracks in the period of interest (winter 2019) were qualitatively assessed and only tracks with a clear signal from the (sea ice) surface were chosen. Several of the tracks were completely disturbed by clouds. If these tracks were included in the analysis most of the cloud contaminated photons would most likely be discarded by selecting only those assigned as high confidence, because the on-board processor would already have identified the photons to be of low or medium signal confidence. This will be discussed in further detail in Sect. 3.3.1. We reiterate, that the signal confidence flag does not discriminate between reflections from the sea ice/ocean surface and from clear reflections originating from e.g., the top of clouds. It is merely an flag providing confidence of whether the on-board processor has identified the photon as an actual signal photon, and not a background photon coming from e.g. the Sun. To eliminate sea level changes and differences between actual sea surface height and the geoid, we subtract the mean elevation of each segment from all of the individual photon elevations within that segment.

We then examine the distribution of the highest elevation taken relative to the mean elevation of each segment. This relative elevation or elevation anomaly, $h_a$, of the highest elevation within a segment of $i = 150$ photons subtracted the mean elevation of this segment, is defined as:

$$h_a = [h_{max} - h_{mean}]_{150}^{i=1}, \qquad (1)$$

where $h_{max}$ is the highest elevation within a given segment and $h_{mean}$ is the average elevation of the same segment. Thus, we find one $h_a$ for each IS2 segment. To investigate whether the elevation anomalies correspond with sea ice features visible in SAR frames, we compare the elevation anomalies from 27 March 2019 with C-band SAR imagery from S1, see Fig. 2 (acquired ~11 hours after IS2 acquisition, see Table 1).

From Fig. 2, it is clear that small elevation anomalies (~0.3m or less) appear over fast ice regions (northern-most part of the track, dark purple, region 2A), and that some higher elevation anomalies, likely caused by land contamination, appear in this data (southern-most part of the track, bright colours, region 2B). We note, that no land mask has been applied to remove photons covering land. In the drift ice region far from the coast differences in elevation anomalies are evident. In the southern part of the track, an area associated with higher degree of deformed ice (visible as high backscatter in SAR data, bright colours) yields higher IS2 elevation anomalies (green-blue-purple colours) than was found over fast ice. Furthermore, edges of the drifting ice crushing against fast ice can also be seen in the elevation anomalies (northern part of the track, bright colours, region 2C) showing the increased deformation likely to occur where the drifting sea ice are pushed against the stationary, fast ice.

When we examine the distributions of the elevation anomalies, we find that the distribution of the relative elevations differ with respect to the different FIS DIR zones (Fig. 3a), but with significant overlap. However, if we select only the highest 20 %, 10 %, 5 % or 1 % of the relative elevations (Fig. 3b–e) falling within a DIR zone, the separation between the DIR zones

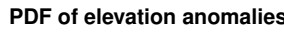

**Figure 3.** Distributions of elevation anomaly ($h_a$) to estimate different degree of ice ridging (DIR) zones. DIR2, DIR3 and DIR4 are shown in green, blue and red, respectively. **(a)** distributions of $h_a$ of all segments, **(b)** distributions of 80th percentile values of $N = 150$ photon segments (the 20% highest values), **(c)** the 90th percentile values (10% highest values), **(d)** 95th percentile values (5% highest values), and **(e)** 99th percentile values (1% highest values). Modal value and number of included observations (n) for each distribution are provided in the graphs.

increases significantly. Focusing only at the high end of the distributions (1–10% of the highest relative elevations) corresponds to investigating segments where there is significant height difference between the highest measurement and the mean elevation – that is, focusing only on the highest ridge sails within a segment. This suggests that DIR in our study area can be estimated
from ATL03, simply by looking at the highest percentile of relative elevations.

**Table 2.** Estimated mode, mean absolute deviation (MAD) and intervals (thresholds) from 95th percentile (5 % highest values) of the elevation anomalies, $h_a$, for each DIR zone. The thresholds are given by the modal value $\pm$ MAD. Adjusted intervals are simply to exclude the gap between DIR3 and DIR4. Values in parenthesis show the elevation anomaly $h_a$ using the 98th percentile, instead of the highest elevation, subtracted the mean (excludes $\sim$3 observations for each 150 photon segment), possibly useful for excluding possible noise measurements. The 98th percentile approach is discussed further in Sect. 3.3.

| Ridging zone | Mode (m) | MAD (m) | Intervals (modal $\pm$ MAD) (m) | Adjusted intervals (m) |
|---|---|---|---|---|
| DIR2 | 0.43 | 0.05 | 0.39 – 0.48 | 0.38 – 0.48 |
|  | (0.33) | (0.05) | (0.28 – 0.37) | (0.28 – 0.37) |
| DIR3 | 0.54 | 0.06 | 0.48 – 0.60 | 0.48 – 0.60 |
|  | (0.42) | (0.05) | (0.37 – 0.47) | (0.37 – 0.49) |
| DIR4 | 0.69 | 0.06 | 0.63 – 0.75 | 0.60 – 0.75 |
|  | (0.55) | (0.05) | (0.49 – 0.59) | (0.49 – 0.59) |

We built a simple threshold-based classification scheme to extract DIR values from IS2, based on the distributions in Fig. 3. While there is a clear separation already at 90 %, large overlaps remain between the distributions for DIR2 and DIR3 (Fig. 3c). This precludes the use of a simple threshold based classification to distinguish between the two DIR zones in this case. For the 95 % percentile (i.e. the 5 % highest elevations) the overlap between the DIR zones is reduced (Fig. 3d). Since the distributions are skewed towards higher elevations, we use the mean absolute deviation (MAD) to estimate the thresholds instead of the classic standard deviation (SD). MAD is simply the median of the absolute deviations from the median and acts as a more robust dispersion/scale measure in the presence of outliers, whereas SD is especially affected by outliers (Leys et al., 2013). Estimations of mode, MAD and the given intervals for each DIR zone (modal $\pm$ MAD) using the 95th percentile data, are provided in Table 2. To exclude a gap between DIR3 and DIR4, a small adjustment (of $\leq$ 0.03 m), based on manual interpretation, is applied to the intervals. We note, that the intervals are based on the 5% highest elevation anomalies (using 95th percentile data), as we assume the highest elevation anomalies will include information on the ridges. Were one to use all of the elevation anomalies, it would also include elevation anomalies from the level ice (as seen by the overlapping distributions in Fig. 3a).

## 3 Results and discussion

### 3.1 Degree of ice ridging in the Bothnian Bay

By using the simple threshold-based classification method, we classify IS2's geolocated photon heights into the different DIR categories and present the results in Fig. 4, together with the DIR zones provided by the FIS ice charts. As expected, IS2 photons classified as DIR2 (slightly ridged ice) occur in all FIS DIR zones simply because there are areas with smoother

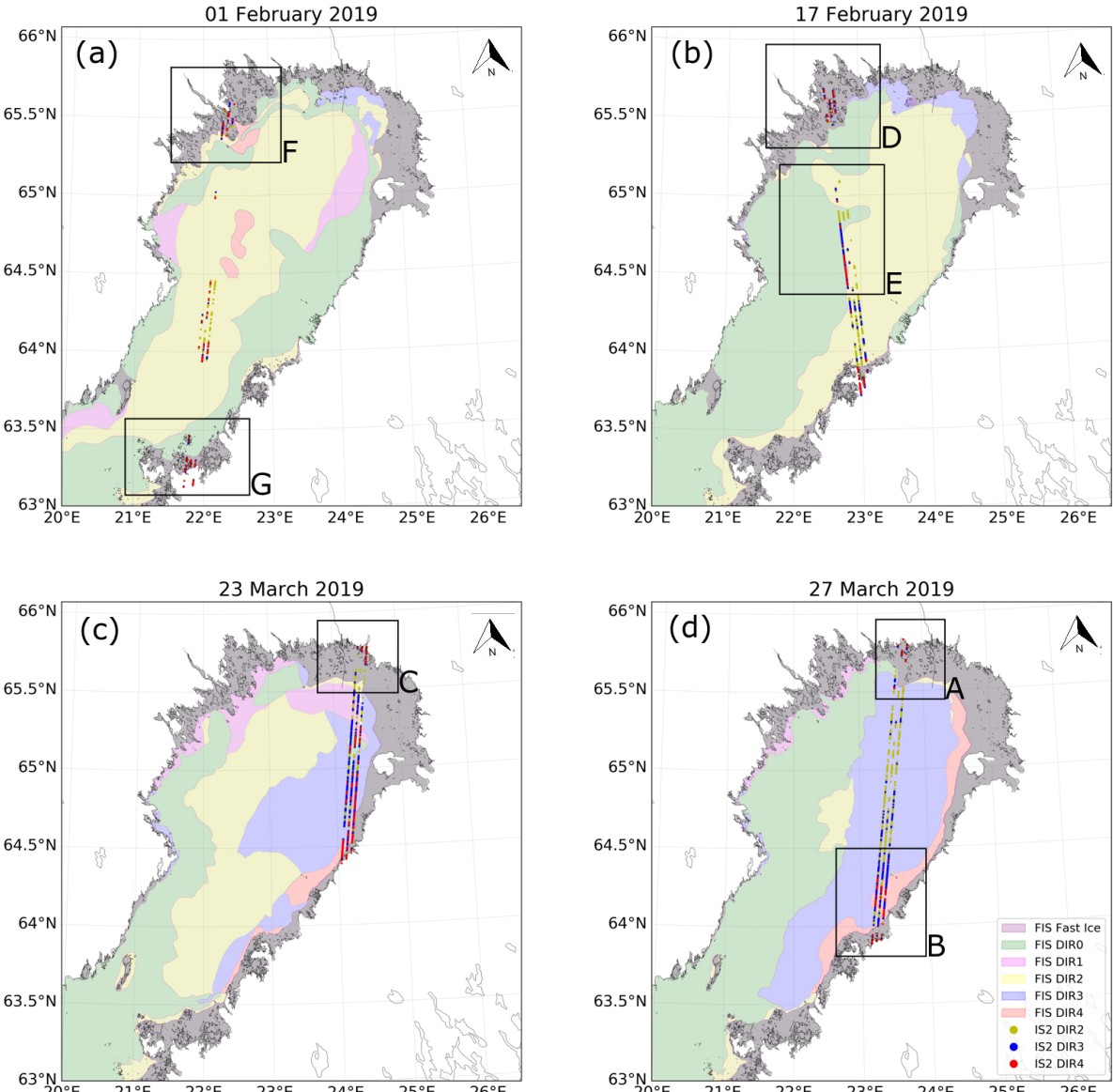

**Figure 4.** Categorised DIR data over four available days (strong beams); **(a)** 1 February, **(b)** 17 February, **(c)** 23 March and **(d)** 27 March 2019. Contours show DIR polygons derived from Finnish Ice Service (FIS) ice charts. IS2 photon heights have only been classified for DIR2–DIR4, where several photon heights could be extracted from. Darker areas close to the coast denotes fast ice. A–G denotes different areas of interest mentioned in the text.

surfaces, i.e. level ice floes between ridges, in all zones. Similarly, IS2 derived values of DIR3 and DIR4 will also be present in a DIR2 zone since even if the area has comparably little deformation, there may be individual ridges present. In other words,

IS2 is able to distinguish features at much smaller scales than the resolution of an ice chart or indeed what is practical for tactical navigation.

The general behaviour of the distributions of IS2 DIR estimates follows the DIR zones from the ice charts. However, IS2 data carries much more information than just the overall DIR for the zone. As mentioned before, the ice chart DIR is a simplification, and in reality large areas that have been assigned to one single DIR are a mixture of several ice types and stages of deformation. If ridge features are sparsely distributed and the area has a relatively large amount of open water, the zone will be assigned DIR2 by the FIS. For heavily deformed ice, Fig. 4d shows a cluster of IS2 classified DIR4 near and in the DIR4 zone of the ice chart (Region B). The presence of IS2 DIR3 is also larger in the southern part of the track (near the border of DIR4) compared to the northern part in line with the ice charts. There is a large amount of IS2 DIR2 values in the DIR3 area suggesting that this part of DIR3 has less ridging. However, this is only based on an amplitude parameter representing the ridge sail heights and one track; the surrounding behaviour and conditions of the ice are not known and it is expected that DIR2 will be present both in the DIR3 and DIR4 zones.

In general, Fig. 4c seems to have more IS2 DIR4 values (even compared with Fig. 4d) that actually encounters a region classified as DIR4. Since this track is close to the coast, more deformation is expected to occur due to the ice drift pushing ice floes towards the coast and fast ice. What is also clear from all four days (Fig. 4a–d), is that when IS2 travels over fast ice regions, the DIR values are almost non-existent except for few segments close to the coast or over small islands (e.g. Region A–D, and F–G), caused by land contamination. We attribute this to regions of fast ice primarily consisting of smooth level ice represented by small differences in the elevation anomalies, and the fact that level ice overall carries more snow than drift ice, which could smooth the surface even further. For the Regions A–D (Fig 4c–d) and G (Fig. 4a) it is clear that there are few DIR values over fast ice and the ones that are detected are actually located over land. Region F (Fig. 4a) is behaving unexpectedly; there are significant DIR values which cannot be explained by land contamination, i.e. these are not close to the coast. Region F has some coastal values, but mostly DIR2 values. This can be partly explained by the fact that it is very close to a DIR2 polygon or that the ice within this zone is slightly deformed. Region D–E (Fig. 4b) shows higher IS2 DIR values (DIR3 and DIR4) in places that are not within or nearby a similar FIS DIR zone. This could be caused by several things; it may be caused by photons within a segment, which are not surface photons but are instead caused by background photons from the Sun. While the histogram approach mentioned in Sect. 2.2.1 keeps photons flagged as surface signal, the bins categorised as surface can still include some background photon events (if the background rate threshold is set too low), thus introducing a spread about the true surface. Some of the photons may also be caused by a (low-lying) cloud cover interfering with the photons, which has not been entirely filtered out in the high signal confidence pre-processing step. Furthermore, these could be proper measurements of the ice surface affected by ocean waves penetrating the sea ice, causing an increase in the surface roughness (specifically near Region E), but also land contamination could explain this (e.g. Region A–D, and F–G). Thus, in order to automate the estimation of IS2 DIR values to use for daily ice charting purposes, it would be necessary to look into additional pre-processing steps to exclude the photons disturbed by cloud-cover, ocean waves and land contamination, as these can affect the result. Once identified, these events could be assigned a measure of confidence to warn the ice analyst. Such flag could also be applied to the DIR values affected by background photon events.

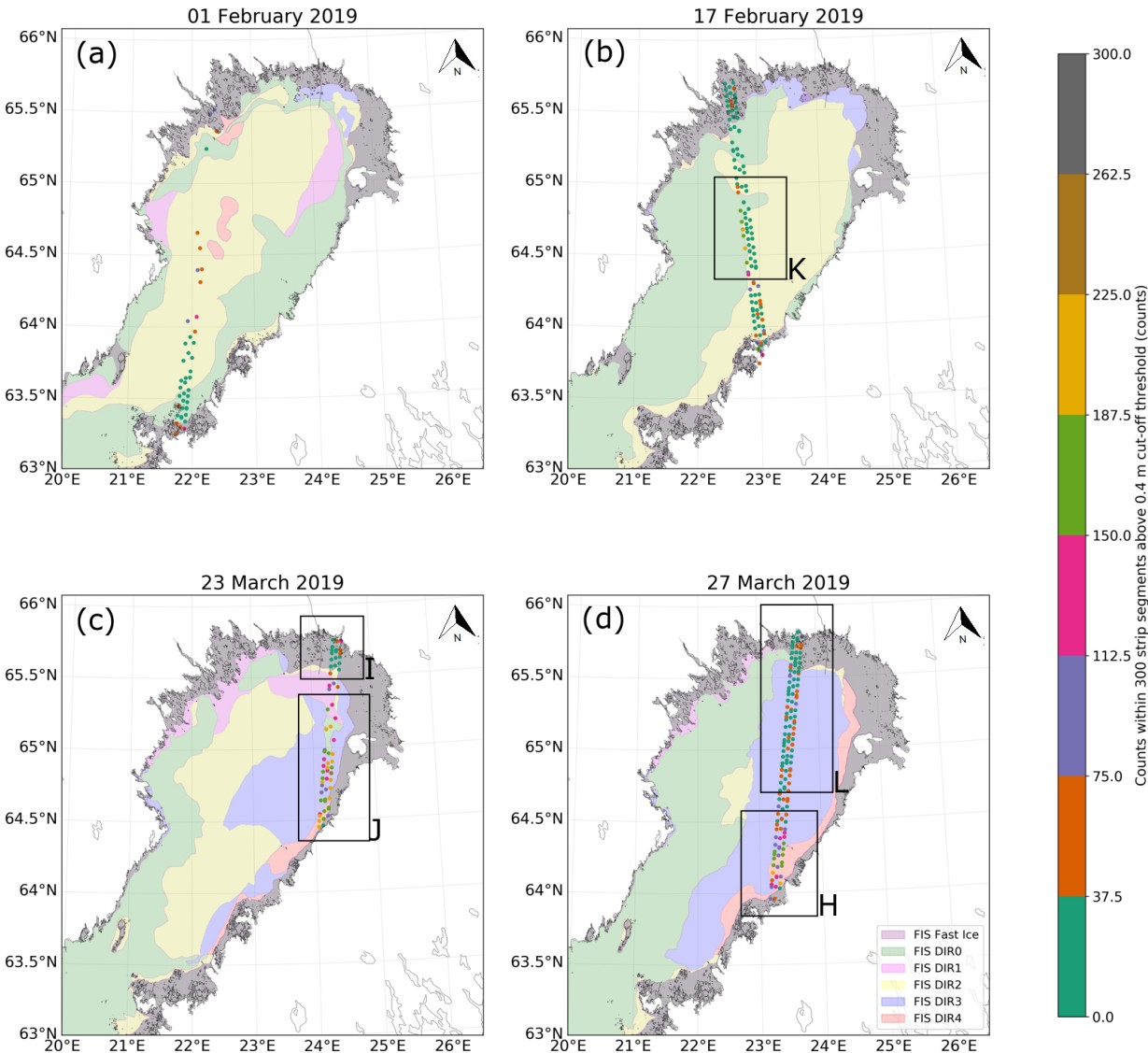

**Figure 5.** Counts of sails, within aggregated strips of 300 elevation anomalies, with values exceeding 0.4 m, acquired over four available days (strong beams); **(a)** 1 February, **(b)** 17 February, **(c)** 23 March and **(d)** 27 March 2019. Contours show DIR polygons derived from FIS ice charts. Darker areas close to the coast denotes fast ice. Colour-bar denotes amount of elevation anomalies, $h_a$, within a strip of 300 elevation anomalies exceeding 0.4 m (cut-off height). H-L denotes different areas of interest mentioned in the text.

Furthermore, although DIR is not uniformly defined and is highly dependent on the ice analysts interpretation of the provided auxiliary data, such as icebreaker routine reports and SAR imagery, it is widely accepted that DIR is dependent on both amplitude (ridge sails) and spatial distribution of ridges (ridge density). The results presented in Fig. 4 focus on the amplitude

of the sea ice roughness, and categorise DIR based on different amplitudes of ridge sail heights. However, the density of ridges

is even more important to consider when navigating in ice-covered waters, since the higher ridge density, the harder it will be to navigate. To investigate the spatial distribution of ridges, we aggregate the elevation anomalies into strips of 300, thus covering an along-track distance of ~5 km (assuming an average distance between the elevation anomalies, $h_a$, of 17 m). For each of the strips, we count how many of the elevation anomalies exceed a cut-off height of 0.4 m, which would indicate a ridge in ice conditions encountered in the Baltic Sea (Lewis et al., 1993). The distribution of the number of elevation anomalies

above the cut-off height within a strip of 300 elevation anomalies is provided in Fig. 5. The small number of counts (green and light purple colours, e.g. most of the track in Fig. 5a) represent areas associated with a low ridge density, which is to be expected over lower DIR areas. Regions of particular interest are denoted by letters H–K in Fig. 5. In Region H, the higher ridge densities coincide with the highest DIR area, i.e. DIR4. The count for one strip in Region H is around 75–225 out of the 300 elevation anomalies, which (by using a value of 180 counts per ~5 km), is equivalent with 36 ridges per km. In the

helicopter-borne electromagnetic (HEM) study utilised in Gegiuc et al. (2018) acquired primarily in March 2011, they obtained measurements in the Eastern Baltic that on average provided a ridge density for DIR4 of 21.5 ridges per km. Hence, the ridge densities for DIR4 obtained by IS2 are higher compared to the average of the HEM study of Gegiuc et al. (2018). However, the ridge densities for IS2 DIR4 are expected, since the HEM study did encounter ridge densities in the range of 0–50 ridges per $1 \times 1$ nautical miles (NM) cells (Gegiuc et al., 2018).

Region I shows very low count values, which is expected as the track covers fast-ice regions. This is also visible as low amplitudes (low ridge sail heights) in the same area (Region C) in Fig. 4. In Region J, there is a high spatial density of ridges with high sail heights (Fig. 4c) even though, according to the ice charts, the track does not coincide with DIR4 areas, i.e. the lowest DIR area the track coincides with is DIR3. However, the count values of Fig. 5c is even higher than in Region H.

This can be explained by the measured sea ice being located close to the coast and/or the fast ice region, and the typical drift

pattern in the Bay of Bothnia causing the sea ice to be pushed towards the eastern part of the Bay of Bothnia (near the island of Hailuoto, see Fig. 1 for precise location). Thus, the deformation of sea ice will be rougher – and most likely result in higher sails (Fig. 4b) – which will explain the higher counts in Fig. 5c. Finally, Region K shows a part of the track that suddenly experiences high counts over low DIR areas, similar to the amplitude values in Fig. 4b. As has already been mentioned, this may be due to unfiltered photons from background sun events or cloud cover, i.e. noise photons. The effect of ocean waves of

significant height (above the cut-off height) has not been investigated here, but should not be neglected, as studies have shown how IS2 photons can identify waves (e.g., Klotz et al., 2020; Horvat et al., 2020). We do acknowledge that it is not unlikely, that ocean waves will occur in this area and could be above the cut-off height, as the area (on the left of the yellow polygon in Fig. 5b) is classified as *very open ice* in the FIS ice chart, and the impact should be investigated in future studies.

Region L (Fig. 5d) represents one of the longest transects over a high DIR zone (DIR3), however this is not well reflected in

the associated ridge densities, where the counts are less than 100 and more often less than 50. It is also seen, in Fig. 4d, how IS2 DIR2 was the prominent DIR zone here, whereas FIS had identified it as DIR3. This could be caused by, e.g. a relative small variation in the elevation anomalies of the sea ice cover, even though it was identified as DIR3 by the ice analyst from the analysis of SAR images. We reiterate, that the ice charts has a lower resolution compared to IS2, meaning that areas with high

**Table 3.** Overview of the investigation of cases. *cases based on proxy for ridge densities, in Fig. 5.

| Region | Date | FIS DIR | IS2 DIR or *counts | Photon profile | SAR image |
|--------|------|---------|--------------------|----------------|-----------|
| A | 27/03 | Fast ice | DIR4/– | – | – |
| B | 27/03 | Fast ice/ DIR3–4 | DIR3–4 | P1 | SI |
| C | 23/03 | Fast ice | DIR4/– | – | – |
| D | 17/02 | Fast ice | DIR4/– | – | – |
| E | 17/02 | DIR1–2 | DIR2–4 | P4 | SIII |
| F | 01/02 | Fast ice | DIR4/– | – | – |
| G | 01/02 | Fast ice | DIR4/– | – | – |
| H* | 27/03 | DIR3–4 | 75–225 | – | – |
| I* | 23/03 | Fast ice | 0–75 | – | – |
| J* | 23/03 | DIR0–1/3 | 75–262.5 | P3 | SII |
| K* | 17/02 | DIR1–2 | 0–225 | – | – |
| L* | 27/03 | DIR3 | 0–75 | P2 | SI |

FIS DIR zones cannot necessarily be expected to have the same DIR occurring everywhere in the associated IS2 observations
due to the different resolutions. We will further investigate these unexpected cases in Sect. 3.2.

### 3.2  Comparison of SAR and IS2 data

The cases to be studied here, highlighted also in Fig. 4-5, are listed in Table 3, with the respective specifications of IS2 and FIS
DIR (or IS2 ridge densities/counts).

Regions A, C–D, F–G, and I are all regions classified as fast ice in the FIS ice charts. Hence, there should be no DIR values
identified by IS2 (Fig. 4) or very few counts (0–50, Fig. 5). This is in line with our results, as presented in Table 3. For some
of these regions, IS2 do encounter DIR4 values due to land contamination (Fig. 4). As we already pointed out in Sect. 3.1,
these outliers could be removed with a detailed land mask, and we will not investigate these cases further here. Region B and
H represent the same area, and are essentially observing the same behaviour (high FIS DIR, high IS2 DIR, high IS2 ridge
densities/counts), thus we will only look at this region with focus on IS2 DIR (region B), and not on the IS2 proxy of ridge
density. Region E and K also represent the same areas and experience the similar behaviour, however the behaviour is not as
expected (low FIS DIR, high IS2 DIR, high IS2 ridge densities/counts). Since IS2 DIR and ridge densities are both observing
the same behaviour, we will only look at the region from the perspective of IS2 DIR (region E).

Thus, apart from the cases causing differences due to land contamination (A, C–D, F–G, I) the results yield four cases to be
further investigated, i.e. regions B, E, J, and L. For this purpose, each of the cases will be compared with a near-coincident SAR

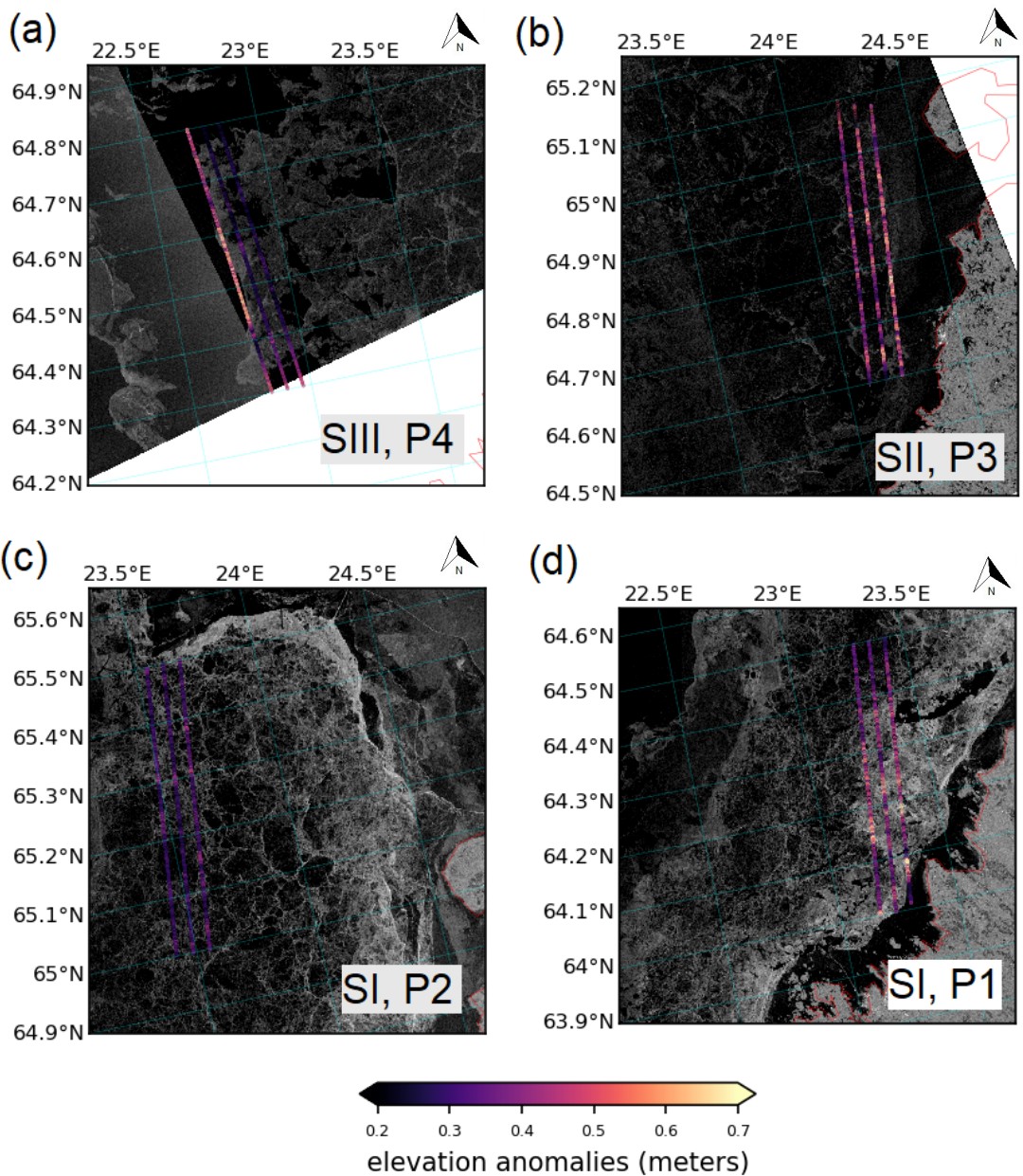

**Figure 6.** Elevation anomalies overlaid S1 SAR image (SI-SII in HV polarisation) for four cases (see Table 1 and 3 for specifications; the figures are representing **(a)** case for region E on 17 February 2019, **(b)** case for region J on 23 March 2019, **(c)** case for region L on 27 March 2019 and **(d)** case for region B on 27 March 2019, P1-P4 highlights the relative photon profiles to be found in Fig. 7). Strong backscatter (white areas in SAR image) is usually perceived as deformed ice in HV polarisation.

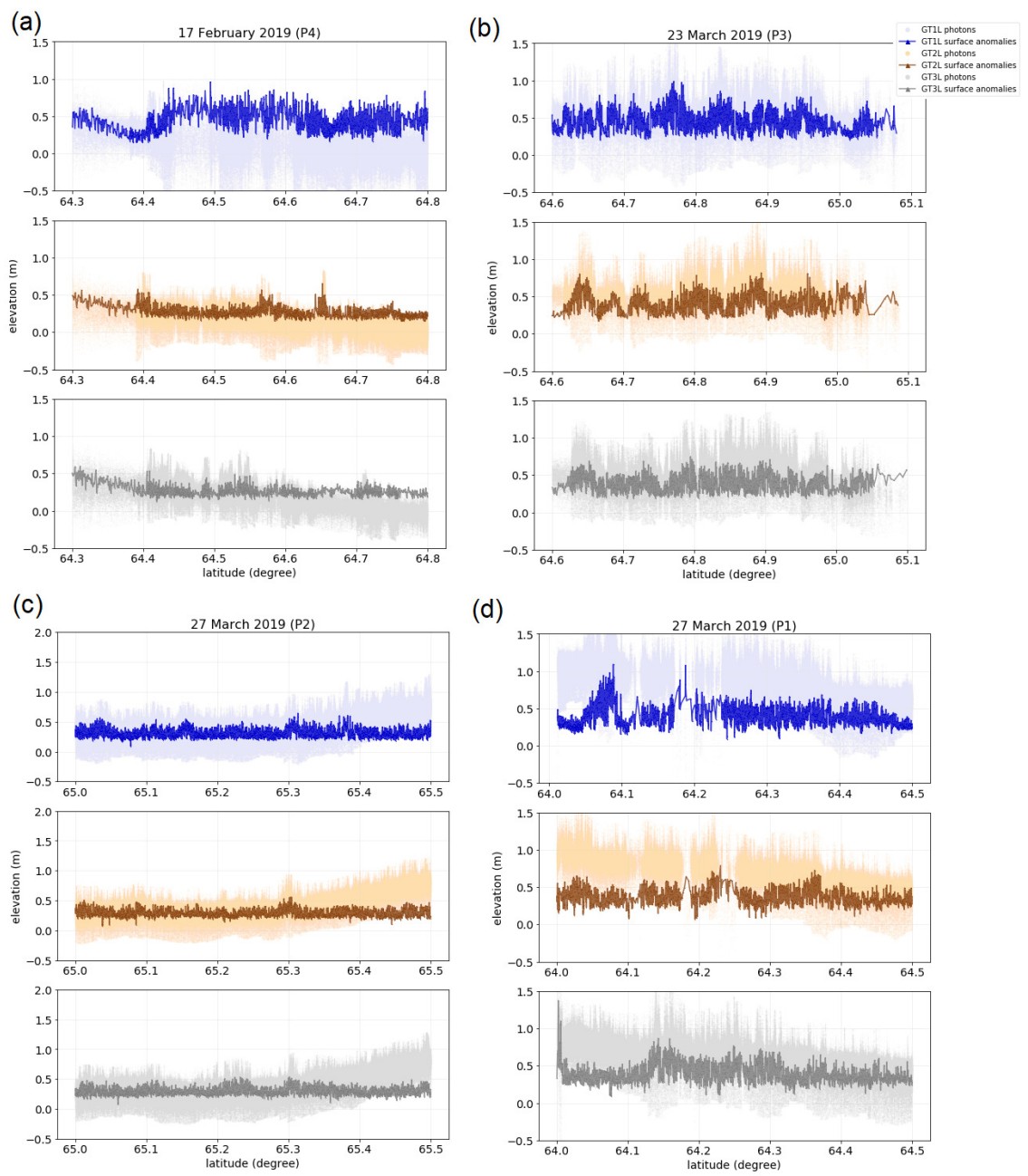

**Figure 7.** Photon profiles (P1-P4) of strong beams ATL03 photons (separated into each strong beam; GT1L, GT2L, GT3L) with elevation anomalies overlaid for four cases (see Table 1 and 3 for specifications; the figures are representing **(a)** case for region E on 17 February 2019, **(b)** case for region J on 23 March 2019, **(c)** case for region L on 27 March 2019 and **(d)** case for region B on 27 March 2019).

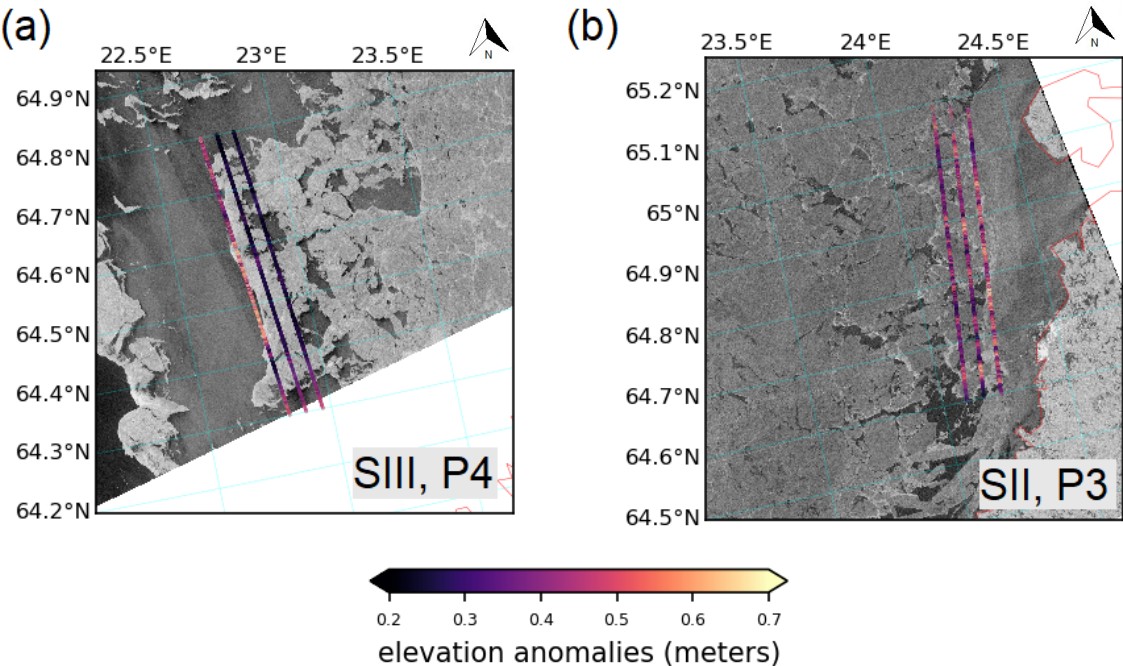

**Figure 8. (a)** S1 C-band SAR image (SIII, HH polarisation) acquired 17 February 2019 with elevation anomalies from IS2 overlaid, 8 hrs 5 m between acquisitions. Similar to Fig. 6a, but with different SAR polarisation. **(b)** S1 C-band SAR image (SII, HH polarisation) acquired 23 March 2019 with elevation anomalies from IS2 overlaid, 2 hrs 26 m between acquisitions. Similar to Fig. 6b, but with different SAR polarisation.

image (SI-SIII) and their individual photon profile with elevation anomalies (P1-P4), see Table 3 and Fig. 6-7. The SAR frames presented in Fig. 6 are all provided in HV polarisation, as this tend to be the most optimal combination to resolve deformation features. Some of the SAR frames for region SII and SIII, are also presented in HH polarisation (Fig 8) to better discriminate between ice and open water/thin ice. In general, high backscatter (bright colours) corresponds to deformed ice (rough ice) in HV SAR frames and low backscatter (dark colours) corresponds to open water/thin ice in HH SAR frames.

Region E on 17 February 2019 covering FIS DIR1–2, but identified as DIR2-4 in IS2 has been investigated in more detail. The IS2 DIR within FIS DIR2 zone is also identified as DIR2 in IS2. However, on the border of FIS DIR1-DIR2, the elevation anomalies from IS2 has been identified as DIR3–4, which warrants further investigation. If we look at the photon profiles (P4) in Fig. 7a, we see that the high elevation anomalies stems from a large amount of subsurface scattering (high density subsurface photon cloud) in strong beam GT1L. In the ATL03 'Known Issues Document' (https://nsidc.org/sites/nsidc.org/files/

technical-references/ICESat2_ATL03_Known_Issues_v003_Aug2020.pdf, accessed 18 February 2021), this phenomenon has been attributed to multiple scattering which can occur over surfaces with e.g., heavy blowing snow or dense fog. This results in a widening of the surface return with more photons erroneously occurring from under the actual surface. We have retrieved

observations on wind speed in the Bay of Bothnia from the weather station at Kemi (Ajos), located at 65.67°N, 24.52°E, provided by FMI (https://en.ilmatieteenlaitos.fi/download-observations, last accessed 18 February 2021). The wind speed was 8 m/s on average at the weather station on 17 February 2019, temperature well below zero and relative moisture content 90 %. In the hours (04:00-10:00) around IS2 acquisition time (07:44:19, see Table 1), the average wind speed was 10 m/s. This could indicate the subsurface scattering to be caused by blowing snow or Arctic steam fog over open water, but there is no way to confirm this. Potentially, the IS2 DIR identified as DIR3–4 could also stem from ocean waves (as suggested in Sect. 3.1). To investigate what surface type is observed by IS2 around time of acquisition, e.g., low ice concentration/open water compared to ice cover, we have extracted a SAR frame for comparison (SIII in HV and HH polarisation in Fig. 6a and 7a, respectively). From the SAR frame, it is clear that over the ice, small variation in the IS2 elevation anomalies are observed over less deformed ice (low backscatter), and areas over more deformed ice (higher backscatter) are experiencing an increase in elevation anomalies. The beam-track (GT1L) is measuring, what in the SAR image appears dark, where the most of the high elevation anomalies occur. We emphasise, that the subsurface scattering/multiple scattering likely caused by heavy blowing snow has not been accounted for in this methodology yet, and will affect the results if the density of the subsurface photons is high, which will affect the overall average value of an aggregated 150 photon segment. It has been notified as a 'Known Issue' in the ATL03 data product, and should be mitigated, should one aim to use the elevation anomalies (or IS2 DIR classifications presented here) for ice charting purposes.

Region J has been investigated with photon profile (P3) and by comparison with SAR imagery (SII), see Fig. 6b and 7b. This region experienced differences in FIS DIR zones (DIR0–1 and DIR3), but many of the elevation anomalies detected by IS2 identified DIR3–4 in this region. In particular, over FIS DIR0, IS2 DIR3 was identified (Fig. 4). In SII (Fig. 6b), we do see a correlation between high backscatter (bright colours) and higher elevation anomalies. Furthermore, the photon profile (P3, Fig. 7b) does not appear to experience a lot of subsurface returns (multiple scattering), as was the case for Region E. Here we also use HH polarisation SAR data, see Fig. 8b. From Fig. 8b, it can be seen that for areas of low backscatter, the elevation anomalies are lower.

Region L on the 27 March 2019, covering FIS DIR3, that encountered very few "ridge counts" (or few IS2 DIR above DIR2) warrants further investigation. A photon profile (P2) has been extracted, and the elevation anomalies is compared with S1 image (SI) (Fig. 6c and 7c). Note here, that the photon cloud (P2) is biased high for latitudes larger than 65.3°N, caused by the differences between the actual sea surface height and the geoid. This bias is eliminated in the pre-processing step as described in Sect. 2.4, and will not affect the final elevation anomalies. While FIS DIR states it to be DIR3, both the IS2 elevation anomalies and the SAR image (SI) do not indicate significant ridging and/or deformation (high backscatter ∼ white areas). The photon profile generally does not have significant multiple subsurface scattering, and the few higher elevation anomalies identified follows the photons identified at high elevations. Generally, it indicates that IS2 DIR is following the actual local ice distribution, and that the DIR category from FIS is based on the larger-scale sea ice conditions and not the small-scale roughness as observed by SAR (SI) and the IS2 elevation anomalies (Fig. 6c and 7c).

Region B on the 27 March 2019, which covered both fast ice and DIR3-4 in FIS ice chart, was identified as DIR3-4 in IS2 DIR, and is compared with S1 image (SI) and the photon profile (P1), see Table 3 and Fig. 6d and 7d. Here, a correlation

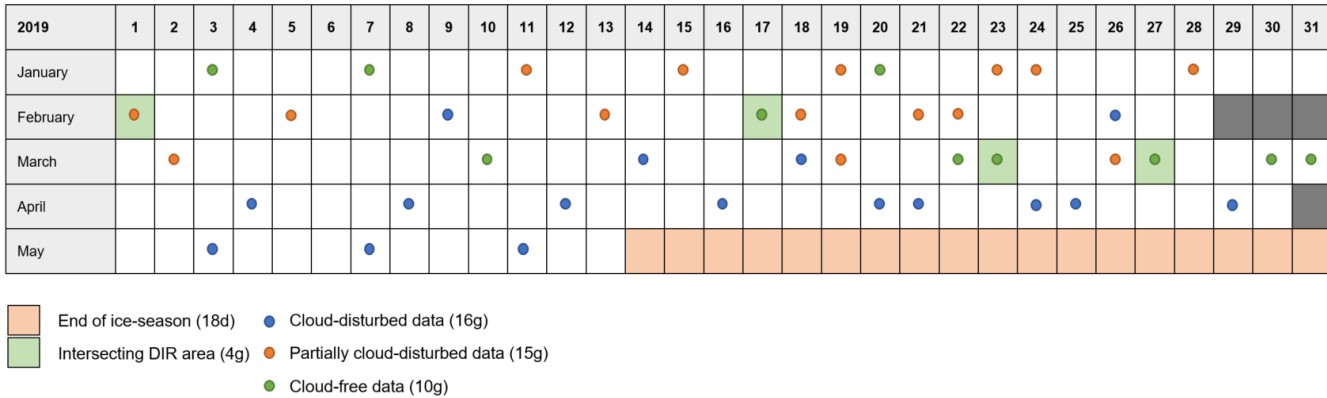

**Figure 9.** Availability of IS2 data granules (tracks) available during spring 2019 (January–May). Each dot represents an available granule which, by visual inspection, is categorised as either: cloud-disturbed data (blue), partially cloud-disturbed data (orange), or cloud-free data (green). Numbers in parenthesis denotes number of granules (g) or days (d). Coloured cells denote either end of ice-season (orange), intersecting DIR area (green) or not applicable (grey).

between high backscatter (white areas) with high elevation anomalies (bright colours) is seen. Similarly, over the fast ice region (dark area/low backscatter) near the coast, lower elevation anomalies are observed. Generally, here the elevation anomalies follow the surface roughness observed in the SAR image (SI) with rough deformed areas (large backscatter) having higher elevation anomalies.

### 3.3 Implications for IS2 and future work

#### 3.3.1 Impact of cloud-cover

Our study is limited, as at time of submission, IS2 data for 2019–2020 were not completely available yet. Even if it were, 2019–2020 was an exceptionally mild winter in our test area, and we would expect to find very few usable overpasses of IS2 over ridged ice. Throughout January to March 2019, only 25 granules of data passing the Bothnian Bay were available (which were either not or only partly affected by clouds to be removed by pre-processing steps, determined by visual inspection). During April–May (until 14th of May where the ice season ended according to BIM (2019)), only 12 granules were available, however all affected by cloud cover. Of the 25 granules between January–March, 10 of these were considered 'cloud-free' and 15 considered 'partially cloud-disturbed' (Fig. 9), determined by visual inspection. Of the 25 granules, only 4 granules intersected a FIS DIR area to such an extent, that a decent amount of photons could be extracted and used in this analysis. A harsher ice season would experience more ice formation and deformation, likely causing more granules to intersect with DIR areas. It should be noted, that while the data are limited, the four granules intersecting a DIR area presented in this study, were measured both during the beginning of the ice season, where the more weaker and less deformed ice occurred (February, where only DIR2 is registered by the FIS ice charts), and over the more heavily deformed ice (March, where both DIR3 and

DIR4 are registered in the FIS ice charts). From Fig. 9 it can be noted, that during January–March it was possible to retrieve data that were either cloud-free or only partially cloud-disturbed, whereas in April–May, no data were possible to retrieve due to persistent cloud cover. Thus, the end of the ice season and ridging hereof could not be properly investigated. January had surface photons available for all available granules intersecting the Bay of Bothnia, however the deformation of the ice cover was not extensive enough at that time for the ice analysts to claim an area as slightly ridged (DIR2) or higher, thus not of particular interest to this study. While IS2 data are impacted a great deal by cloud cover, this study was still able to investigate ridging occurring during the beginning of the ice season, with mild deformation, and towards the middle/end of the ice season, with the highest deformation occurring. Therefore, we assume the granules used in this study to represent the ice deformation occurring in the Bay of Bothnia well, albeit during a mild ice season.

The Baltic sea is relatively small and located South of 66°N, chosen for this study due to the availability of quality reference data. Nonetheless, this study shows the potential of IS2 data to supplement other sources of information for ice charting. Furthermore, with the expected increase of shipping in the Arctic as a response to the continuing melting of the sea ice cover (e.g., Melia et al., 2016), these results prove valuable not only for the Baltic ice services and ice charting, but also for international ice charting. With the increased IS2 coverage in the Arctic, compared to the Baltic Sea and Bay of Bothnia, more photon elevations and tracks become available, therefore we do not foresee cloud-cover as much of a restriction in Arctic ridge determination during the winter period (Oct–Apr). Increase in cloudiness in the Arctic has been reported in spring (Mar–May) (Schweiger, 2004), suggesting that spring and likely also summer months will experience an increase in loss of data due to increased amount of cloud cover. The same is observed for the Baltic, in Fig. 9, with no data available from April and on wards the end of the ice season. Even with increase in clouds during spring and summer, recent studies have shown the first steps in using IS2 observations to estimate melt ponds in the Arctic summer sea ice (Tilling et al., 2020), showcasing that some valuable information can be retrieved by IS2 in the Arctic during summer. Possibly, even information on summer sea ice deformation can be retrieved along-side melt ponds.

With the cloud cover impacting the photon product, there could be a great benefit to apply a cloud flag in the pre-processing stage to remove erroneous points originating from clouds. This yields the question: is there a potential for applying cloud flags in the processing? The ATL03 product does not provide a cloud flag, nor does the ATL07 (surface heights) or ATL10 (freeboard) products. Indeed, only the atmosphere product (ATL09) provides a cloud flag, however as stated by Kwok et al. (2021), the resolution of the cloud flags available in ATL09 is too coarse to provide useful filtering (at least at the lead segment scale, ∼27–40 m). The cloud flags are sampled every ∼400 m (Kwok et al., 2021), hence the resolution of the cloud flags will also be too coarse to use over deformed sea ice (determined by DIR from elevation anomalies, calculated on average every ∼17 m). Should the ATL03 in the future be provided with quality flags useful to remove photons impacted by clouds, this would have great benefit to methods using the lower-level ATL03 photon data product, such as presented in this study.

### 3.3.2 Ridge anisotropy in the Bay of Bothnia

The ice charts do not include the orientation of ridges. As IS2 data are transect data by nature, isotropy of ridge orientation would complicate the comparison. The ridge orientation has been studied by the FIS in the past, and the conclusion was that in

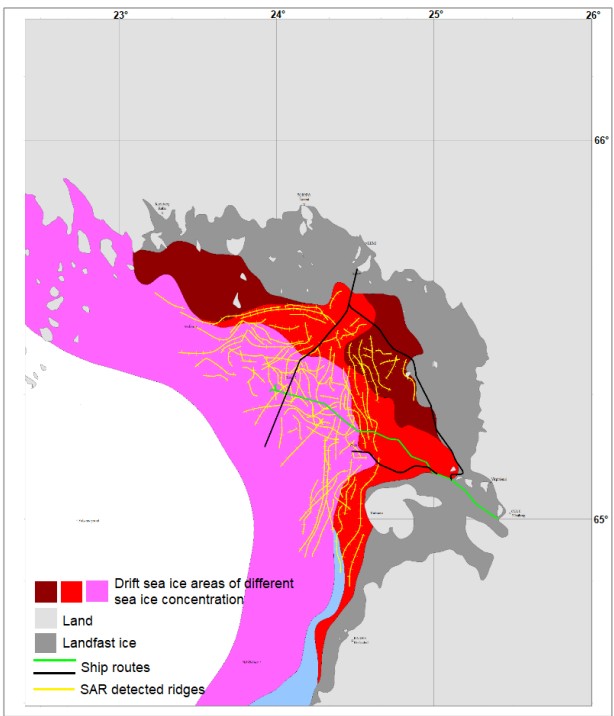

**Figure 10.** Ridges close to the ports of Oulu and Kemi, winter 2013-2014. Courtesy of FIS/Finnish Transport Infrastructure Agency. Note that only the ridges falling within the area of ship channels (black and green lines) were analysed.

the vicinity of the land fast ice the preferred orientation of ridges is the local direction of the coastline. However, further away the ridge orientation is anisotropic (see Fig. 10). Most of the IS2 data used in this study are located far from the zone where orientation isotropy is found. Thus, our study is not significantly affected by ridge orientation. However, in future studies, the presence of preferred ridge orientation could be detected from difference in ridge densities between ascending and descending orbits. The orbit in direction of the preferred ridge orientation should measure lower ridge density than the one perpendicular

to it. However, an absence of cloud cover would be necessary to ensure that both ascending and descending tracks are available without cloud-contaminated photons.

### 3.3.3 Ice floes and impact in low ice concentration areas

Potentially, a higher elevation anomaly could be due to sudden changes of surface; the drop in elevation between ice floes and sea surface. We reiterate, that the elevation anomalies are based on the difference between the highest elevation within a

395 segment and the average value of that particular segment. In that regard, if the segment of 150 photons occurs over a change of surface types (ice floes vs open water), potentially the highest elevation anomaly could originate from the top of the ice floe (where snow could also introduce additional scattering, potentially increasing the elevation of the photon), and the average value of the segment could be influenced by photons from the sea surface, which generally have lower elevations compared

to ice floes. Thus, the impact of low ice concentration areas, as the surfaces here will range between ice floes and sea surface more often, warrant more investigation, although is out of the scope of this study. Potentially, one could argue that the ridge identification and sea ice deformation information retrievable from IS2 should only apply in areas where sea ice concentration exceeds 70%, which is a common threshold applied an altimetry-derived sea ice thickness due to the difficulties of retrieval of sea ice freeboard over areas of low sea ice concentration. However, this also suggests that a combination of IS2 data (potentially the elevation anomalies or IS2 DIR identification scheme presented here) and SAR images can provide interesting and important information for the ice analysts. SAR images can provide the overall sea ice distribution and deformation, whereas IS2 can be used over particular deformed areas to identify the impact of the ridging by providing the height of the ridge sails. Potentially, combining the data with high resolution optical images could help identify which surface IS2 encounters as well. A study by Petty et al. (2021) have linked Sentinel-2 (S2) optical imagery with IS2 (ATL07/ATL10) to estimate e.g., chord length of the ice floes. Hence, combining ATL03 with S2 (and/or Sentinel-3) optical imagery could also provide useful insight on surface classifications, and optical images are also interesting in combination with IS2 due to the impact of cloud cover that affects both laser and optical acquisitions.

### 3.3.4  Impact of noise photons in the used methodology

Throughout this paper, we work under the assumption that the highest photon elevations within a segment originate from a ridge sail. There is a possibility that they actually originate from a background sunlight event, fog, blowing snow or cloud cover. Nonetheless, we find the overall elevation anomalies correspond with expected sail heights in the area, and it therefore seems likely that the confidence flag in the ATL03 product is of very good quality.

To evaluate the impact of possible noise photons being the highest measured elevation within a 150 photon segments, we investigated percentiles of the segments with mean value subtracted from each of the elevations, rather than the highest elevation subtracted the mean elevation (the elevation anomaly, $h_a$). Thresholds based on the 98th percentile of the mean subtracted the photon elevations rather than $h_a$ (see Table 2) have similar intervals (about 0.10 m smaller of each interval), however the distribution of the classified DIR zones are similar to those of the elevation anomalies (as presented in Fig. 4). This suggests, that even by using only high-confidence photons and the highest value, in most cases the photons do in fact originate from the top of the ridges. Hence, for ridge sails less than 1 m in height, the on-board filtration scheme of low, medium and high-confidence photons seems to keep only surface photons including photons originating from the top of the ridge sails, and successfully removes noise photons. Using the 98th percentile instead of $h_a$ excludes approximately 3 photons per 150 photon segments, thus due to dead time and the single photon-counting method of IS2 (Neumann et al., 2019b), the top of the ridge may not be included if the highest photon of the segment is excluded. However, the 98th percentile thresholds are also within range of average ridge sail heights determined by former studies (of 0.5–0.6 m, see e.g, Lewis et al., 1993; Gegiuc et al., 2018). Further validation of ridge sails and surface measurements of small-scale roughness, such as ridges in the Baltic Sea, observed by IS2 by comparing with airborne observations is encouraged.

Furthermore, it should be noted that this study utilises version 2 (v2) of ATL03 product (Neumann et al., 2019a). ATL03 was upgraded to version 3 (v3) in May 2020, but due to reprocessing and processing lags, there might be gaps in the v3 data.

Hence, we have kept using v2 data, especially since the updates of v3 do not seem to impact our data significantly. Four major updates were presented in v3 (see https://nsidc.org/data/atl03, Version Summary):

v3_1   A quality assessment parameter indicating the percentage of reference photons within a certain distance from the reference digital elevation model (DEM) has been introduced,

v3_2   Photons that do not intersect the DEM height within a +/- 30 m buffer are effectively removed from consideration by the signal classification processing. Furthermore, photons poorly geolocated (provided by a new flag, *podppd_flag*) are no longer classified as potential signal,

v3_3   A logic error in the code combining the precise orbit determination (POD) and precise pointing determination (PPD) degrade values has been fixed, and a flag (*podppd_flag*) has been introduced,

v3_4   Finally, two new parameters indicating nearly (*near_sat_frac*) or fully (*full_sat_frac*) saturated ATLAS shots are introduced.

v3_1–v3_2 will generally affect the amount of photons to be removed in the pre-processing stage (+/- 3 m from the geoid
elevation), however will not have a significant impact on the results, i.e. the signal photons should not be removed by the +/- 30 m from DEM buffer introduced in v3_2. The *podppd_flag* (v3_3) could potentially remove badly geolocated photons, that are still within the +/- 3 m from geoid elevation buffer that is applied in the pre-processing step of this study. However, we assume this to be of small significance. If a few photons within the photon cloud are badly located, it will have a small impact on the average value (used to calculate elevation anomalies), and if a badly geolocated photon turns out to be the highest value within
a segment (used in calculating the elevation anomaly), it could potentially be mitigated by using e.g., 98th percentile instead of the highest photon (as described in the beginning of Sect. 3.3.4). Potentially, the two new parameters (explaining near and full saturation of ATLAS, v3_4) could be used to further process the data (e.g., mitigate the the effect of a first-photon bias that the photon-counting lidar is affected by, see e.g. Sect. 7.7.3 in Neumann et al. (2020), usually applied to higher level products). When ATLAS is fully saturated, the surface return photons will show a gap in height with no reported photons for the duration
of the detector dead time. However, it is stated in Neumann et al. (2020), the cases where near or full saturation of ATLAS proves difficult and require special handling are mostly related to higher-level products (ATL03+). For future work, v3 will be used and potentially future quality assessment indicators could be included to aid the pre-processing and remove erroneous point measurements e.g., photons originating from clouds or multiple surface scattering, should a future quality indicator on this aspect be provided in the ATL03 product. Nonetheless, we found the v2 of ATL03 used in this study to be of high quality
already.

### 3.3.5   Uncertainty of IS2 DIR

Currently, the IS2 DIR intervals are based on the selection of elevation anomalies originating from different FIS DIR areas, and at a later stage the computed IS2 DIR are compared to the same FIS DIR. However, the resolution of FIS DIR is quite coarse, thus a direct validation in terms of e.g., a confusion matrix, that would show exactly how many IS2 DIR were correctly/falsely

classified within FIS DIR polygons, is to some extent useless. This is also why we do not aim to say that this comparison with the FIS DIR constitutes as a real validation, but rather as a comparison with the polygons used to identify different IS2 DIR intervals. However, this raises the question of whether training data by the same polygons also proves useless due to the coarse resolution. IS2 provides sub-scale information (compared to FIS DIR) and by comparing with SAR images, we can identify if the classifications of IS2 DIR (based on elevation anomalies) actually follow the ridging and deformation pattern of the Baltic. This will indicate whether training with FIS DIR for IS2 DIR seems feasible, as the FIS DIR are based partly on icebreaker observations and partly on SAR images. Based on comparison with S1 SAR images, we show that the elevation anomalies correspond to surface roughness (e.g., expected ridge sails in the Baltic) and that the surface information available from IS2 is detailed (high elevation anomalies identified over high backscatter in HV SAR images, see e.g., Fig. 6d, which usually is an indicator of sea ice deformation). However, since the classification of this study is based only on the 5% highest elevation anomalies, either more FIS DIR data or quantitative S1 data should be included and used for training, to ensure a robust classification.

However, this raises a different point: the potential of doing a quantitative validation with S1. For now, we have done a qualitative validation by comparing the elevation anomalies with S1 SAR images. Potentially, a quantitative validation could be made with both the elevation anomalies and the identified IS2 DIR with the S1 SAR images, to see if this qualitative relationship can be validated quantitatively. However, for a comprehensive quantitative validation with S1, it would be necessary to (1) correct the observations for the sea ice drift occurring between S1 and IS2 acquisitions ($\sim$0.10–0.45d, see Table 1), and (2) take into account the different spatial resolutions, so that the resolution of the IS2 DIR/elevation anomalies fit with the resolution of S1. Drift products in the Baltic are limited to buoy data (sparsely distributed in the Baltic, but available in the the Bay of Bothnia, e.g., Karvonen, 2012) and SAR-based drift products (Karvonen, 2012; Karvonen et al., 2020), however the quality of the SAR-based product is limited. The drift is given only between different time instants corresponding to the SAR acquisition times for each of the overlapping SAR pairs. Furthermore, there are several areas where no data exist due to the algorithm not being able to reliably detect ice drift everywhere. Furthermore, there is some error in the ice drift estimation in the current algorithm, as it tends to underestimate the ice drift magnitude (personal communication, Karvonen, 2020). The limitations of the ice drift product and re-sampling into same resolution makes the comparison between S1 and IS2 additional work, that is out of the scope of this study.

However, even with the limitations described above, a large amount of ice information is retrievable from IS2 photons, as indicated by this study, and more work should be invested into utilising IS2 data to more than just conventional freeboard-to-thickness estimations. Our goal was to show how the IS2 data performs over different ice regimes, in addition to the regions used to train the algorithm, and illustrate how IS2 data might augment the information in the ice charts. By comparison with S1 SAR images, we have shown how small-scale roughness information is available from IS2 – which is otherwise non-retrievable from other altimeters which footprints and sampling frequencies do not allow for the high density surface sampling that IS2 does. Furthermore, the comparisons of IS2 and S1 have shown how it is possible to identify deformation occurring over different ice regimes by differences in the magnitude of elevation anomalies. Finally, we have presented one method that

this IS2 small-scale roughness information could be converted into something of use for ice navigators (IS2 DIR) – albeit more work is necessary to refine this methodology, as has been discussed throughout Sect. 3.

## 3.4 Future outlook

We have emphasised the need for a near-real-time or fast delivery photon product before (Sect. 3.1), but it should be mentioned that even without time critical products, it would still be valuable to estimate the ice conditions in certain areas for planning purpose. An example of such planning would be compiling the statistical information on ice conditions required by the International Maritime Organisation (IMO) polar code to create a regional climatology (IMO, 2020). IS2 would then be an independent source of ridging estimates in the ships planned operations area.

Currently, the data latency of the ATL03 data product is 21 days (Brown et al., 2016). Should a near-real-time or a fast delivery photon product become available, IS2 data could be used by operational ice services. Information on ice surface elevation is of high priority, not only to support safe ice navigation in the Bay of Bothnia, but also across the Arctic. Due to along-track nature of IS2 measurements, they are most useful when combined with image type data. The main information of ice services at the moment is satellite SAR imagery. However, because there are several processes contributing to SAR backscatter, estimating the ice type from SAR data is necessarily an ambiguous process. Areas of high backscatter are often interpreted as strongly deformed areas. Alas, high backscatter can also result from centimetre scale surface roughness (Manninen, 1997), for example frost flowers that are of the magnitude of SAR wavelength on top of thin new ice. These two properties of ice, similar in SAR data, are different from the viewpoint of ice navigation. For FIS, regular reports from ice breakers are used to support the analysis of SAR frames. In areas where these reports are not available, IS2 would provide a potentially valuable estimate of ridging.

In addition to data latency, usability of ICESat-2 in operational ice charting is also affected by data availability. Measurements are limited to the cloud free portions of the orbit pattern, and an in-depth explanation of data availability in our study can be found in Sect. 3.3.1. However, we emphasise that in many remote areas where in situ observations are not available, the IS2 based estimates of ridging may be the only option independent of SAR frames.

## 4 Conclusions

In this study, we have presented the correspondence between FIS DIR, satellite SAR data and elevation anomalies using geolocated photons heights (ATL03) measured by IS2 during spring 2019 in the Bay of Bothnia. DIR derived from IS2 using our methodology follows the general expectations in the Baltic. This is the first time the feasibility of IS2 is studied from the viewpoint of winter navigation and operational ice charting. DIR is one of the most important parameters used in ice navigation as it indicates whether or not a vessel can safely pass through an ice-covered area. In the Baltic, daily ice charts provided by the FIS include information on DIR based primarily on in situ ice breaker observations and partly derived from SAR imagery.

Furthermore, we find that in some cases (in particular three out of five highlighted areas) along-track densities of relative elevations (elevation anomaly, $h_a$) above a threshold cut-off height of 0.4 m are consistent with the distributions of the FIS

DIR areas. Heavy deformation is found on sea ice close to the coast/fast ice regions. This is expected due to the sea ice drift pattern in the Baltic pushing ice floes towards the coast causing deformation. Typical ridge densities and sail heights expected in the Bay of Bothnia correspond well to the elevation anomalies.

In addition, this study demonstrates how much surface topography information of small-scale roughness ($<$1m) is measured by IS2 and kept, even when applying the high-confidence flag of the on-board filtering scheme. Thus, even over the thin sea ice areas of the Baltic Sea, one can benefit from the high density surface sampling and information that IS2 provides. Compared with S1 SAR images, the elevation anomalies follow the deformation observed in the SAR images. We note, that this methodology of estimating IS2 DIR is based on the highest 5% of the elevation anomalies, and that while the IS2 DIR follows the deformation seen in the SAR frames (qualitative assessment), the IS2 DIR only follows FIS DIR for some cases.

To develop a reliable DIR level classification algorithm from IS2, more FIS DIR observations or quantitative SAR data should be included. We conclude that a time-critical IS2 product would be of benefit to ice services around the world complementing widely-used satellite SAR data.

*Code and data availability.* All data can be obtained by contacting the first author. ATL03 products were retrieved from NASA's Earth Search at https://search.earthdata.nasa.gov/, using version 2 (Neumann et al., 2019a). S1 frames were processed by ESA and available from

545 the Alaska Satellite Facility at https://search.asf.alaska.edu/#/. Ice charts were provided by FIS. Shapefiles to produce map of Baltic Sea (Fig. 1) were retrieved from HELCOM, and are available for download at: http://maps.helcom.fi/website/mapservice/. Scripts used in the data analysis are available online at the following GitHub repository: https://github.com/reneefredensborg/DIR-from-IS2 (https://doi.org/10.5281/zenodo.4636435).

*Author contributions.* RFH, ER and HS conceived the concept of the study. Processing and computing was done by RFH. RFH, ER and

550 HS jointly analysed the data, and SF joined in on discussions in relation to these analyses. RFH wrote the initial manuscript and all authors contributed to the editing of the text.

*Competing interests.* The authors declare that they have no competing interests.

*Acknowledgements.* We thank FIS for providing their ice charts and expert advice, the Polar Oceanography and Sea Ice group at the Finnish Meteorological Institute for a great discussion and for interesting questions relevant to this study, and Kyle Duncan for useful insights on

using IS2 for ridge detection in the Arctic. We thank the two anonymous reviewers and our editor for valuable comments that improved the manuscript.

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
