# Peer review of "Estimation of degree of sea ice ridging in the Bay of Bothnia based on geolocated photon heights from ICESat-2"

_The Cryosphere, 2020_

## Referee Comment (RC1) · Anonymous Referee #1 · 10 Nov 2020

**Review of: Estimation of degree of sea ice ridging in the Bay of Bothnia based on geolocated photon heights from ICESat-2**

**1 General Comments**

**1.1 Paper Synopsis**

I found this to be an interesting paper on a topic of significant human relevance. In it, the authors use the ICESat-2 laser altimeter to retrieve the ridginess of ice in a way comparable to that traditionally charted by ice analysts for marine navigators. I believe their analysis to be generally rigorous, and the figures in particular were well presented. I have relatively few concerns about subsequent publication of this manuscript, although I would like my questions concerning ridge-alignment anisotropy and cloud-cover to be directly addressed before publication.

**1.2 Reproducibility**

It is unfortunate that the FIS charts are not publicly available, and so this analysis is not replicable without contacting the authors. I do however appreciate that this is not within the authors' control if FIS insist on holding the copyright. It was pleasing to see the analysis code uploaded to an open repository, with nice use of markdown for context. Although this is beyond the remit of my role as reviewer, I would encourage the authors to ensure that all defined functions have docstrings so as to boost code readability. I would also encourage the authors to consider storing their code in a persistent location such as that provided by Zenodo, for which it will receive a digital object identifier (DOI).

**2 Specific Comments**

L126: "We also discarded all measurements that deviated from the geoid elevation by more than 3 m". I feel like the reader would benefit from knowing what fraction of photons this is? Is there a risk that you're throwing out more photons during high tides than low tides? What's the typical sea surface anomaly on the geoid here? Wouldn't you be better off discarding measurements based on their deviation from the tide-corrected sea surface?

Fig 2: what are the units of the y axis? I appreciate that because it's a histogram, 'Density' corresponds to the number of relevant photons per bin, but what's the bin-width? Without knowing that, the specific values on the y axis are meaningless. Just an idea, but consider displaying this data as a probability density function (i.e. plot the probability of the max-height photon having a given elevation anomaly). By doing that you could perhaps squeeze in some meaningful information on the y axis rather than it just being 'density'.

L211: You talk about the drift pattern in the Bay of Bothnia being West to East. I imagine a consistent drift-pattern like this would produce an anisotropy in ridge orientation (probably aligning them along the North-South axis?). Because IS2 tracks also run broadly North-South, it seems to me you might have a sampling issue here, particularly with regard to ridge density. To restate this, you're only measuring ridge-density along one direction, and that's (generally) the direction that the ridges themselves might preferentially run. I feel this needs to be addressed in your discussion. Are ridges aligned anistropically in your study area? Can you use the fact that IS2 has non-parallel ascending and descending tracks to get a handle on this?

This paper argues that IS2 DIR retrieval would be of significant operational benefit to ice analysts, but does not substantially address the impact of cloud cover on measurement availability (although it does discuss the role of low lying clouds on surface ranging biases). The authors state that only 25 granules of data were available in the three-month period that they searched within. Firstly, I feel that a reader that is non-expert in IS2 would benefit from the term 'granule' being defined here. Second, while this number is of some use, I think the reader would benefit more from knowing on what number of days can the study-area be analysed for DIR in this way? Were those 25 granules confined to a small cluster of days, or do the granules represent the three month time period well? Are there some months where DIR is more retrievable than others due to cloud masking? These questions significantly affect the utility of the proposed retrieval method.

**3 Technical Points**

L83: it seems unnecessary to define an acronym (RF) for random forest when that acronym is not subsequently used.

L83: I think you mean dual-polarization and not dual-frequency (my understanding is that RADARSAT is just C-band).

L181: should by "low-lying" I think

L196: would perhaps be clearer if phrased "The distribution of the *number* of elevation anomalies". Or perhaps *frequency* is the right word? I feel that 'amount' is a bit vague for a quantity that is presumably well defined. Again in 197: the small *number* of counts.

L199: "the most interesting regions..." this seems like a subjective statement (what is the *most* interesting) worded as fact. I guess they're interesting to you because of the variability or magnitude of ridge density/amplitude. Perhaps rephrase slightly to "Regions of particular interest are..."

L204: Grammar: "the ridge densities for DIR5... *are* higher compared to...". Technical Point: You're also saying two things in this sentence: (a) you make a valid and useful comparison between HEM and IS2 ridge density (b) you make an unsupported statement ("however not unlikely") about the feasibility of your retrieved IS2 ridge density. I would separate these two claims into distinct sentences, and I think you should offer a citation to support the reasonable-ness of your higher densities. If this isn't available to you then I would avoid the statement that it's 'not unlikely'.

L207: Not sure a "high spatial distribution" is meaningful. Do you mean a high spatial density?

L209: coincide**s**

L210: "Once more, this can be described by the sea ice measured ... is close to the coast and/or fast-ice" Needs rewording for clarity.

L214: experience**s**

L224: Even if it **were**

L228: **comparatively**

L232: Do you mean you've emphasised it before in this manuscript? Or in previous works? Clarify.

L237: **sunlight** (one word), the second half of this sentence needs a look at too.

L279: 'tremendous' is quite a loaded and subjective term, consider rewording.

---

## Author Comment (AC2) · 17 Nov 2020

In Fig 1 in AC1: Note that only the ridges falling within the area of ship channels (black and green lines) were analysed.
* * *

---

## Referee Comment (RC2) · Anonymous Referee #2 · 11 Jan 2021

Please see my review in the attached PDF

Please also note the supplement to this comment:
https://tc.copernicus.org/preprints/tc-2020-315/tc-2020-315-RC2-supplement.pdf

---

## Author Response (AR1)

**Editors comments**
Our comments (after 1st review)

Dear Dr. Hansen and co-authors,

I have received two reviews of your manuscript "Estimation of degree of sea ice ridging in the Bay of Bothnia based on geolocated photon heights from ICESat-2." One reviewer recommends minor revisions and I believe all these concerns are straightforward to address. However, the second reviewer has major concerns with respect to training data and the lack of quantitative validation of the degree of ice ridging. I agree with Reviewer 2's comments that additional information and analysis with respect to validation is required. Please submit a revised version of your manuscript addressing the concerns of Reviewer 1 and the quantitative validation concerns raised by Reviewer 2.

Thank you for submitting your work to The Cryosphere.

Dear Editor,

We are grateful for the constructive comments provided by the reviewers. We have taken into account the well-founded concerns from Reviewer#1 and #2, and provide here, point-by-point response to the reviewers' comments (the original response, and additional comments after reviewing manuscript). Furthermore, the revised manuscript and the tracked-changes document are also included. The suggestions and comments from the reviewers have improved the manuscript significantly, and we hope, that you find their concerns to be satisfactorily addressed.

In particular we have added:

- A thorough and comprehensive analysis of several 'critical cases' (that is, cases that either do or do not correspond to the ice chart degree of ridging (FIS DIR) in the expected manner). In this analysis, we have included the photon profiles with the elevation anomalies overlaid, and furthermore also included Sentinel-1 images with the elevation anomalies overlaid to investigate the particular inconsistencies and differences observed in the elevation anomalies.
- A thorough description of the impact of cloud cover, in particular for our test area/study, but also in general on how this will impact the utility of IS2 for ice charting purposes.
- A short section discussing the impact of ridge anisotropy in the Bay of Bothnia, and provided a potential method to mitigate the effect of this, in case the effect is profound.
- A section discussing the impact of ice floe edges/low ice concentration areas, and how this might affect the results, and how to potentially avoid/mitigate this.
- A section discussing the uncertainty of this methodology, in particular the feasibility of training using FIS DIR, and discussing why this study has been limited to qualitative comparisons.
- More examples on how IS2 can be used for ice charting purposes.

- We have included the particular technical corrections requested by the reviewers, and we have made some corrections throughout the text to aid readability, and provide clarifications.

Sincerely, on behalf of the authors,

Renée Mie Fredensborg Hansen

**Reviewers #1 comments**
Our response (first comments)
Our response (second comments)

**1 General Comments**
**1.1 Paper Synopsis**
I found this to be an interesting paper on a topic of significant human relevance. In it, the authors use the ICESat-2 laser altimeter to retrieve the ridginess of ice in a way comparable to that traditionally charted by ice analysts for marine navigators. I believe their analysis to be generally rigorous, and the figures in particular were well presented. I have relatively few concerns about subsequent publication of this manuscript, although I would like my questions concerning ridge-alignment anisotropy and cloud-cover to be directly addressed before publication.

We would like to thank the reviewer for the positive and insightful comments. Please find the initial response to the comments.

Thank you for a thoughtful and indeed helpful review. In particular the questions concerning ridge-alignment anisotropy and cloud-cover, which we have addressed here. This has improved the manuscript.

**1.2 Reproducibility**
It is unfortunate that the FIS charts are not publicly available, and so this analysis is not replicable without contacting the authors. I do however appreciate that this is not within the authors' control if FIS insist on holding the copyright. It was pleasing to see the analysis code uploaded to an open repository, with nice use of markdown for context. Although this is beyond the remit of my role as reviewer, I would encourage the authors to ensure that all defined functions have docstrings so as to boost code readability. I would also encourage the authors to consider storing their code in a persistent location such as that provided by Zenodo, for which it will receive a digital object identifier (DOI).

It is a very good thing the reviewer pointed this out. The Finnish Meteorological Institute has an open data policy, and, by definition, products like FIS ice charts should be publicly available. The sea ice concentration and thickness from the ice charts indeed is available through the Copernicus Marine Environment Monitoring Services (product SEAICE_BAL_SEAICE_L4_NRT_OBSERVATIONS_011_004). However, the Degree of Ridging is not included yet. Thus we have started the internal process at FMI to make them available, and the response from FIS has been very positive. We hope that the DIR will be publicly available as SIGRID3 and NetCDF files by the time we complete the revisions to our manuscript.

Currently, the FIS ice charts are not publicly available (only the thickness and ice concentrations are available as netCDF at CMEMS). We have included a link to the available data products, but have also stated that the FIS ice charts are not yet fully available.

We will ensure that all defined functions have docstrings to boost code readability and will also store the code in a persistent location, for it to receive a DOI. This will be achieved by the time we revise the manuscript.

Additional docstrings have been included, and the code has received a DOI.

**2 Specific Comments**

L126: "We also discarded all measurements that deviated from the geoid elevation by more than 3 m". I feel like the reader would benefit from knowing what fraction of photons this is? Is there a risk that you're throwing out more photons during high tides than low tides? What's the typical sea surface anomaly on the geoid here? Wouldn't you be better off discarding measurements based on their deviation from the tide-corrected sea surface?

We will add the fraction information in the revised manuscript.

Fraction information has been included in the revised manuscript in Sect. 2.4.

We appreciate the reviewer's comment on this, but will highlight that in the Baltic Sea (and in particular in the Bay of Bothnia), the effect of tides is negligent. Overall, in the Baltic, the maximum tide measured by tide gauges has ranged between 18-23 cm (Medvedev et al., 2013, 2016), but within areas not part of the south-western Baltic, tidal amplitude observations have not exceeded 10 cm (Lilover, 2012).

Fig 2: what are the units of the y axis? I appreciate that because it's a histogram, 'Density' corresponds to the number of relevant photons per bin, but what's the bin-width? Without knowing that, the specific values on the y axis are meaningless. Just an idea, but consider displaying this data as a probability density function (i.e. plot the probability of the max-height photon having a given elevation anomaly). By doing that you could perhaps squeeze in some meaningful information on the y axis rather than it just being 'density'.

We will look into if other plots (incl. the one mentioned by the reviewer) might provide additional information beyond the density plot already included.

We changed this to a probability density plot and we included the units.

L211: You talk about the drift pattern in the Bay of Bothnia being West to East. I imagine a consistent drift-pattern like this would produce an anisotropy in ridge orientation (probably aligning them along the North-South axis?). Because IS2 tracks also run broadly North-South, it seems to me you might have a sampling issue here, particularly with regard to ridge density. To restate this, you're only measuring ridge-density along one direction, and that's (generally) the direction that the ridges themselves might preferentially run. I feel this needs to be addressed in your discussion. Are ridges aligned anistropically in your study area? Can you use the fact that IS2 has non-parallel ascending and descending tracks to get a handle on this?

Excellent point. The preferred orientation of ridges has indeed been overlooked in our manuscript! We could, unfortunately, find no peer reviewed literature on the ridge orientation in the Baltic so we asked for help from the FIS. They produced an old study on ridges in our study area, and it is apparent that the ridges tend to form parallel to the fast ice edge - which in turn is parallel to the coastline. See the image below, showing the ridges in Bay of Bothnia during winter 2013-2014. This anisotropy will impact ridge density, which we shall discuss in the revised manuscript. However, as the reviewer points out, this is likely mitigated by the

orbit pattern of ICESat-2. Furthermore, by no means all of the ridges are oriented in the same direction, but especially in areas of heavy ridging the orientation distribution is even (see Fig RC1).

In the spirit of discussion, we would also like to point out that ICESat-2 ridge detection might open possibilities to study the ridge orientation in remote seas. Quite possibly one could see preferential orientation in areas such as the Kara Sea by comparing statistical ridge densities retrieved from ascending and descending orbits! But this, for now, is beyond the scope of the paper at hand.

[Figure]

**Fig RC1:** Ridges close to the ports of Oulu and Kemi, winter 2013-2014. Courtesy of FIS/Finnish Transport Infrastructure Agency. Note that only the ridges falling within the area of ship channels (black and green lines) were analysed.

We have added a new discussion on the topic of ridge anisotropy and its impact both in the Bay of Bothnia and for the IS2 DIR detection in section 3.3.2.

This paper argues that IS2 DIR retrieval would be of significant operational benefit to ice analysts, but does not substantially address the impact of cloud cover on measurement availability (although it does discuss the role of low lying clouds on surface ranging biases). The authors state that only 25 granules of data were available in the three-month period that they searched within. Firstly, I feel that a reader that is non-expert in IS2 would benefit from the term 'granule' being defined here. Second, while this number is of some use, I think the reader would benefit more from knowing on what number of days can the study-area be analysed for DIR in this way? Were those 25 granules confined to a small cluster of days, or

do the granules represent the three month time period well? Are there some months where DIR is more retrievable than others due to cloud masking? These questions significantly affect the utility of the proposed retrieval method.

We will address this in the revised manuscript. During the period of January 2019 - May 14th 2019 (when the ice season ended), a total of 41 granules intersected the Bay of Bothnia, but only 25 were (by visual inspection) considered either partially-impacted by clouds or cloud-free. Of these 25 granules, we found that only four traversed a ridged area during the mild winter of 2019, which happened in February (beginning of the deformation, less ridged areas) and in March (high deformation, heavily ridged areas occur here). In April-May, all granules were completely impacted due to the cloud cover. However, this was not the case in January-March, where granules also were not clustered, but were evenly distributed throughout the months, thus they represent the three-month time period. Hence, we want to emphasize that although there were a high number of useful granules (25), only 4 intersected a DIR area (DIR2-DIR4) within the reference data set that could be used in this study.

We have added a description of what the term 'granule' means with respect to the IS2 data in Section 2.2.

"One orbit track is divided into 14 granules (latitude-dependent region) to limit data size to a maximum of 6 GB, such that the Bay of Bothnia appears in granule region 03 on ascending tracks and granule region 05 on descending tracks (Neumann et al., 2020)."

In regards to the other part of the question, we have added a new figure (Fig 9) to show the distribution of the granules and spread of available granules divided as 'cloud disturbed', 'partially cloud-free' and 'cloud-free' granules. Furthermore, we have highlighted which granules intersected a DIR area (the 4 cases presented in this study). In addition, we have discussed the impact of cloud cover on IS2 (and for operational ice charting) in more detail in Sect. 3.3.1, and in the future outlook section as well.

**3 Technical Points**

All technical points will be taken into account when revising the manuscript.

L83: it seems unnecessary to define an acronym (RF) for random forest when that acronym is not subsequently used. Agreed, removed.

L83: I think you mean dual-polarization and not dual-frequency (my understanding is that RADARSAT is just C-band). Yes, thank you, corrected.

L181: should by "low-lying" I think Yes, corrected.

L196: would perhaps be clearer if phrased "The distribution of the number of elevation anomalies". Or perhaps frequency is the right word? I feel that 'amount' is a bit vague for a quantity that is presumably well defined. Again in 197: the small number of counts. Corrected to 'number' instead of 'amount'.

L199: "the most interesting regions. . . " this seems like a subjective statement (what is the most interesting) worded as fact. I guess they're interesting to you because of the variability or magnitude of ridge density/amplitude. Perhaps rephrase slightly to "Regions of particular interest are..." Yes, agreed, we have corrected it accordingly.

L204: Grammar: "the ridge densities for DIR5. . . are higher compared to. . . ". Technical Point: You're also saying two things in this sentence: (a) you make a valid and useful comparison between HEM and IS2 ridge density (b) you make an unsupported statement ("however not unlikely") about the feasibility of your retrieved IS2 ridge density. I would separate these two claims into distinct sentences, and I think you should offer a citation to support the reasonable-ness of your higher densities. If this isn't available to you then I would avoid the statement that it's 'not unlikely'. The term 'not unlikely' was deemed supported by the ridge densities described in the sentence, before, however we agree that it may not have been entirely clear. For clarity, we have changed the sentence to the following:

"Hence, the ridge densities for DIR4 obtained by IS2 are higher compared to the average of the HEM study of Gegiuc et al. (2018). However, the ridge densities for IS2 DIR4 are expected, since the HEM study did encounter ridge densities in the range of 0–50 ridges per 1×1 nautical mile (NM) cells (Gegiuc et al., 2018)."

L207: Not sure a "high spatial distribution" is meaningful. Do you mean a high spatial density? Yes, has been corrected to 'density'.

L209: coincides Corrected.

L210: "Once more, this can be described by the sea ice measured ... is close to the coast and/or fastice" Needs rewording for clarity. Rephrased to:
"This can be explained by the measured sea ice being located close to the coast and/or the fast ice region, …"

L214: experiences Corrected.

L224: Even if it were Corrected.

L228: comparatively Corrected.

L232: Do you mean you've emphasised it before in this manuscript? Or in previous works? Clarify. Clarified which section this was emphasised in by including the following "(Sect. 3.1)"

L237: sunlight (one word), the second half of this sentence needs a look at too. Thank you, have been rephrased to:
"There is a possibility that they actually origin from a background sunlight event or cloud cover disturbed photons."

L279: 'tremendous' is quite a loaded and subjective term, consider rewording. Changed 'a tremendous benefit' to 'of benefit', to make it less subjective.

**References**

Medvedev, I.P., Rabinovich, A.B. & Kulikov, E.A. Tidal oscillations in the Baltic Sea. Oceanology 53, 526–538 (2013). https://doi.org/10.1134/S0001437013050123

Medvedev IP, Rabinovich AB and Kulikov EA (2016) Tides in Three Enclosed Basins: The Baltic, Black, and Caspian Seas. Front. Mar. Sci. 3:46. doi: 10.3389/fmars.2016.00046

M. -J. Lilover, "Tidal currents as estimated from ADCP measurements in "practically non-tidal" Baltic Sea," 2012 IEEE/OES Baltic International Symposium (BALTIC), Klaipeda, 2012, pp. 1-4, doi: 10.1109/BALTIC.2012.6249181.

**Reviewers #2 comments**
Our response (first comments)
Our response (second comments)

**General comment**
The manuscript addresses the important problem of deriving the degree of sea ice ridging from the ICESat-2 laser altimeter measurements. Data and methodology are well presented, however the accuracy of the results and applicability the method to other ICESat-2 measurements is questionable. As suggested below a major revision is needed to assure feasibility of the presented method.

We thank the reviewer for a thorough and valuable review of our paper! The concern of applicability of the method to other ICESat-2 measurements is not without basis and we hope to address it sufficiently both in this response and in the future revised manuscript.

We thank the reviewer for the valuable review of our paper which indeed has improved the paper, and we hope that we have addressed the concerns satisfactorily.

My first concern is that the same data is used both to train the method (by analyzing the elevation anomalies histograms and setting the intervals in Section 2.3) and to validate the method (by comparison of along-track IS2 DIR with maps of FIS DIR in Section 3.1). Generally speaking, usage of the same data both for training and validation precludes conclusion of extrapolating the algorithm applicability. What if it worked well only on these data points?

The reviewer makes a valid point here, and this is also something that we considered when we prepared the original manuscript. This is also the reason why we kept the comparison of our ICESat-2 estimates and the ice charts qualitative. Were we to have independent ridging estimates, we could do a more complete validation of the product. Initially we planned to use winter 2019-2020 data purely for validation but alas 2019-2020 was a record mild ice season and as such, there is not enough data to perform a robust validation.

There is the possibility of only presenting elevation anomalies across the different DIR regions, however we think the most relevant and important result in our paper is that the elevation anomaly distributions differ for different DIR zones (based on FIS charts) on four non-consecutive days and separate locations (when combining the observations, Figure 2 in preprint) to such an extent that non-overlapping ridging thresholds can be determined for each DIR area, and later applied to the entire track where a general agreement with FIS charts is seen (Figure 3-4 in preprint).

Based on analysis of the critical cases highlighted in Figure 3-4 in the preprint, we have addressed these and discussed in more detail four particular critical cases (cases with diverging results) with both photon profiles (and elevation anomalies), and elevation anomalies overlaid SAR frames, see Sect. 3.2.

My second concern is related to the validation approach and results. Only a qualitative comparison of along-track observations with maps is performed and no quantitative estimation (for example, in a form of a confusion matrix) is provided. Understandingly, the

DIR product from FIS is quite coarse and cannot capture all spatial variations of ice ridging. That makes quantitative comparison with IS DIR less useful. But what if it makes useless also the algorithm training (i.e., finding the elevation anomalies intervals)?

The decision to complete a qualitative comparison between IS2 and the ice charts was made because we recognise the weakness of using the same data for building confidence in our method that was used to train it. As the reviewer correctly points out, this does not constitute a true validation and we do not imply this in the manuscript. Our goal is to show how the IS2 data performs over different ice regimes, in addition to the regions used to train the algorithm, and illustrate how IS2 data might augment the information in the ice charts.

We agree that ice charts are not the perfect comparison data set, but as we point out in the paper (Li. 78-81), they are one of the most reliable sources of information for those operating vessels in the Baltic Sea. The resolution difference between the information in the ice charts and IS2 is several orders of magnitude, and the size of features apparent in the IS2 data is surprisingly detailed. While the users of ice charts may not be interested in individual ridges, even those are captured in the IS2 elevations and this finding is one of the main messages of our paper (Li. 276-279).

Based on the reviewer's comment we see the need to add a clarification to the manuscript regarding our goals and the impact of our findings, and will add this in the revised manuscript.

We have provided in Sect. 3.3.5 a new discussion on training and comparing with the same data, the possibility of quantitatively validating with S1 along with a clarification regarding the goals and impact of our findings.

These two questions are open and I, therefore, suggest changing the emphasis of the paper: instead of presenting an algorithm for sea ice ridging estimation from ICESat-2 (with a questionable training/validation approach) the paper should rather present a comparison of ICESat-2 elevation anomalies with ground truth data. Such a change in the accent implies several major modifications:

• Change of the title, goals and, correspondingly Abstract, Introduction and Conclusions.

We appreciate the reviewers comment on this.

However, we believe that our DIR algorithm is valuable - especially the fact that we present an algorithm in a form that can be easily implemented by anyone with basic control of a programming language. This despite the fact (which we will make crystal clear to the readers in the revised manuscript) that validation of the algorithm using the same data that we used to train the algorithm, tells us little about how well the algorithm compares to other data. This said, it is an easy algorithm to implement and we believe it would be of great benefit to ice services, as a supplement of independent information to their current sources, even if they use it as an invalidated source of auxiliary information. The ice navigation community would greatly benefit from investigations using IS2 high-resolution surface measurements, and this is one of the studies that suggest a possible way of utilising such data. Simply changing the

goals/title/sections to a comparison with ground-truth data, where the conclusions are the same as previously shown, perhaps does not provide any additional or new information, nor the insight we wish to provide for end user services. Nonetheless, we shall do our best to clarify our message in the revised manuscript.

• Plotting of elevation anomalies on Fig. 3 instead of classification into IS2-DIR(1,2,3).

Initially, this was a figure that we did make (however, only for 27 March 2019 when comparing with SAR) and which later led to the DIR estimation. We will add this in the revised manuscript.

This has been added for one day (27 march 2019, see Fig. 2 in revised manuscript), and a more detailed analysis on critical cases has been presented (see Sect. 3.2).

• More detailed analysis of profiles of photon height (and plots of the profiles) in critical cases when FIS DIR doesn't correspond to elevation anomalies.

While we believe that we commented a lot on both positive and negative cases (A-K in Fig. 3 and 4), we understand the reviewers wish to see the profiles. Thus, for a revised manuscript, we will include photon profiles of critical cases and discuss them in more detail.

The critical cases have been investigated in more detail, see Sect. 3.2.

• Comparison of elevation anomalies, or photon height profiles with other independent data, e.g. Sentinel-1, Sentinel-2 in the aforementioned critical cases.

In our initial studies, we did compare SAR images from Sentinel-1 to investigate if the elevation anomalies detected by IS2 were consistent with strong backscatter, which seemed to be the case alas with some exceptions. SAR backscatter is not only a measure of large scale surface roughness, but e.g. volume scattering plays a role as well. Furthermore, SAR images also differ from IS2 in spatial resolution, and also the temporal shift between acquisitions (the SAR frames we collected in the initial studies had more than 0.5 day temporal shift) and sea ice drift will have an effect on the comparison. Finally, the ice charts used in the study are in essence an expert analysis of the SAR frames. At the time of writing, we did not think that such comparison brings much added value to the manuscript.

We can include the comparison of ICESat-2 and Sentinel-1 data. Other information on sea-ice roughness/profiles is limited in the Baltic Sea (and in the IS2 period), thus no direct comparison is made (e.g., comparing with campaign data). However, it should be noted that we have made a comparison in the manuscript already (L. 200-205, on page 11), placing our results in the context of ridge densities found by Geguic et al. (2018) using HEM observations collected in previous years.

Comparison with S1 for critical cases (and for one entire track on 27 march 2019, see Fig. 2) has been presented in Sect. 3.2 and Figure 6 (and Fig 7 and 8 for HH SAR polarisations).

**Specific comments**

Title: Consider "Comparison of degree of sea ice ridging in the Bay of Bothnia with elevation anomalies in geolocated photons from ICESat-2"

We shall revise the paper to satisfy the reviewer's main concerns. However, we do not foresee that this will require a change of the title.

Line 15: "information ON ice conditions"

Line 26. "Divergent motion forms cracks ..., and convergent motion results in …"

Line 91: "from 1 TO 3."

Line 91: "is AN imaginary"

We will include the above mentioned edits, thanks. All four of the above-mentioned edits have been included.

Figure 2. The black line (DIR2) is almost indistinguishable from the dark blue one (DIR3). Consider green, or any other brighter color.

Noted, thank you. This has been changed to green for easier discrimination.

Equation 1. The equation says that you sum up 150 values of h_max and then subtract h_mean. That does not correspond to the text. Equation needs to be corrected.

Thank you - will be corrected. This has been corrected.

Lines 150 – 155. In the first sentence I would recommend to replace "we classify IS2 geolocated photon heights into different DIR categories" with "we compare IS2 geolocated photon height with different DIR categories" and correspondingly rewrite the rest of the paragraph.

Thank you for this comment. However, this would be necessary in case the goal of the paper is changed - for now, since we do not aim to change the goal of the paper, we will keep the sentence.

Figure 3 and 4: Color of fast ice and FIS DIR0 is very close and hard to distinguish. The same applies to FIS DIR 1 and FIS DIR4. Figure caption describes DIR3 as blue and DIR4 as red, but to me it appears as violet and pink.

Yes, that is true. This is simply because there is a transparency in the filled contours due to the fact that we wanted the underlying small islands to be visible in the product, since some areas with elevation anomalies/DIR coincide with or are close to small islands. If the filled contours are not transparent, the islands will not be visible. We will investigate whether the figure can allow the contours to be plotted before the outline of the small islands or if it can be highlighted in some other way. If that is not the case, the transparency will be kept. We will look into alternative colours for DIR0 to see if it becomes more distinguishable.

And yes, the DIR3 and DIR4 does look more violet and pink - again, this is caused by the transparency. However, if we put the exact same colors as contours below, then one cannot see when the IS2 DIR points are on top. We will revise the figure caption, since there is a colour scale available in the Figure that shows the exact colors.

Fast ice color has been changed for easier discrimination. The captions have been changed by removing the description of the colors of DIR, and now only uses the legend provided in the image.

Figure 3 and 4: 27 March appears before 23 March. Is it incorrect title or incorrect map? The same with 01 and 17 February.

Yes, it is in reverse order. We will change that. Thank you. This has been corrected. Letters for specifically mentioned areas (A-L) have been kept in the same order.

Figure 3 caption: "where several photon heights could be extracted from AND compared to …"

This will be changed, thank you. Corrected.

Figure 4: Color of DIR and counts of sails is almost indistinguishable or has very bad contrast (e.g. light blue points on light purple DIR3 polygons).

Yes, we agree. However, we did have issues with choosing a color scale that did not have this problem in one way or another. Thus, we decided to use this one in the end. We will investigate alternative color scales to see if it becomes easier to distinguish. An alternative color scale has been selected, and we hope it will be easier to distinguish by this. We have also combined the color scales of the four subplots.

Line 156. I disagree that behavior of IS2 DIR follows the FIS DIR zones even generally. I would say that they correspond only in 50% of cases. That's why it is better to compare not DIR to DIR but h_a to DIR. This sentence and the paragraph below need to be rewritten accordingly.

We thank the reviewer for this comment. We want to draw the attention to L. 150-160 in the manuscript, where we state:

"Using our method, we classify IS2's geolocated photon heights into the different DIR categories and present the results in Fig. 3, together with the DIR zones provided by the FIS ice charts. As expected, IS2 photons classified as DIR2 (slightly ridged ice) occur in all FIS DIR zones simply because there are areas with smoother surfaces, i.e. level ice floes between ridges, in all zones. Similarly, IS2 derived values of DIR3 and DIR4 will also be seen in a DIR2 zone since even if the area has comparably little deformation, there may be individual ridges present. In other words, IS2 is able to distinguish features at much smaller scales than the resolution of an ice chart or indeed what is practical for tactical navigation. The general behaviour of the distributions of IS2 DIR estimates follows the DIR zones from the ice charts. However, IS2 data carries much more information than just the overall DIR for the zone. As mentioned before, the ice chart DIR is a simplification, and in reality large areas

that have been assigned to one single DIR are a mixture of several ice types. If ridge features are sparsely distributed and the area has a relatively large amount of open water, the zone will be assigned DIR2 by the FIS".

So, it is not reasonable to assume that all points will be labelled the same DIR of IS2 as labelled by FIS. However, we agree that a comparison with SAR images may provide additional information in this area, especially to understand how some areas of FIS DIR3 show as DIR2 in IS2. We plan to include these in the revised manuscript.

We have investigated the critical cases in Sect. 3.2, and have rephrased the paragraph to state that in some cases, the IS2 DIR behavior follows FIS DIR.

Line 167. If "more deformation is expected to occur due to the ice drift pushing ice floes towards the coast" why is that not reflected in FIS DIR? Area west of Oulu is heavily trafficked and presumably the ice charts are the most accurate here as a lot of reports from icebreakers should come. But IS2 reports a lot of DIR4 measurements unlike DIR3 reported by FIS ice chart. Inspection of a SAR image on 23 March 2019 (see below) shows a lot of ice flows separated by leads. Maybe covered with thin ice. Could the IS2 DIR4 be modulated by edges of the floes, rather than the ridges? In my opinion that is a very good example to illustrate better by collocation of IS2 and SAR and showing profiles of geolocated photons and the detected ridge sails.

We thank the reviewer for this expert insight. In fact, this does raise the question of whether the difference between the ice floes and the open water is large enough to create a DIR4 area. If this is the case, the surface anomaly would require a difference between the lead and the top of the ice floe edge to be 0.6-0.75m. The question is then whether the sea ice is this thick in this area. Nevertheless, it definitely requires more discussion, by e.g. studies of ice floe topography using airborne laser scanning could be encouraged (albeit would be in the Arctic, which is a different sea ice regime), or by investigating the topography of ice floes using ICESat-2 - however, that is beyond the scope of this manuscript. A subject that we must discuss here is also the snow on top of the ice floes, as IS2 measures the top of the snow layer. Especially in the shallow area close to Oulu it is not uncommon that pieces of ice are stable for a considerable time and then break up as floes which start to drift. These floes may have a significantly thicker snow layer than floes that have not been stationary, resulting in higher freeboards. In revision, we shall look at the development of ice in this area from a time series of SAR images to gain additional insight on this.

We note that it has not been possible to properly investigate the development of ice in this area from time series of SAR images, as we were not able to find the same SAR frame as presented by the reviewer (thus, we were not properly able to investigate the particular case presented here). Nonetheless, as a critical case, we have investigated the same track (however, more south than shown here), with a SAR frame acquired on the same day (although, this frame did not cover the same area as highlighted by the reviewer). However, looking into the FIS DIR of the following day to investigate potential drift/difference, we now see, that the entire Eastern side of the Bay of Bothnia is considered deformed to a certain degree, and that the open water area shown in the SAR image below (and seen in the FIS DIR for 23 March 2019) is no more. Also, it should be noted, that the FIS DIR for 23 March 2019 takes into account SAR frames acquired before 12.00 noon, whereas the IS2 photons

were acquired at 18:31:14 that evening. Hence, potentially the ice has drifted in the time between the SAR frames and icebreaker reports used to produce the FIS DIR and the IS2 acquisition. See FIS DIR from 24 March 2019 below (does not include the fast ice region, since panoply does not allow for visualisation using two different datasets in one). However, we highlight that areas used to train the IS2 elevation anomalies (so as to estimate IS2 DIR) were specifically that type of DIR area, and did not include e.g., these large open water area which changed FIS DIR by the next day.

[Figure]

FIS DIR

0,0          1,0          2,0          3,0          4,0

Data Min = 0,0, Max = 4,0

However, as a comment to the first sentence here, the deformation increasing near the coast is reflected in the ice charts, both in the one from 23 March, however located further south, and for the 27 March where DIR4 is observed along the entire fast-ice region and near Oulu - the area of where the SAR image is taken. Again, here it must be considered that the winter season of 2019 was mild - and on the 17 February, the sea ice cover did not experience a lot of deformation/roughness beside right above the island of Oulu (DIR3 area). So, it is likely due to the mild winter and sea-ice drift, that you see this opening near the island of Oulu (which is also reflected in the FIS ice charts). However, what you also see is a large DIR3 area near this coast - and DIR4 forming. In fact 4 days later, DIR4 formed along the entire eastern coast of the Bothnian Bay.

[Figure]

Even though we were not able to retrieve the same SAR frame as presented by the reviewer and hence, could not investigate the SAR time series
, we have still included a discussion on the impact of ice floes/low ice concentration in Sect. 3.3.3.

Line 199: On fig 4.a most of green dots (very low number of ridges) occur over DIR3 (ridged ice) which is not discussed in the text. That's a major disagreement which seems to be ignored. For the sake of completes of the study, not only the positive cases but also the negative cases should be highlighted and explained.

Thank you for this observation - it has most definitely not been the intention to ignore this. In fact, we found high variability across cases, that has been discussed in the text (A-G, J-K in Figure 3-4). But, we will comment on this in the revised manuscript, perhaps even include SAR imagery and/or photon profiles to compliment.

We have discussed it in more detail as a critical case in Sect. 3.2.

Line 224: Cloud cover is another major factor limiting availability of IS2 measurements. It cannot be ignored as it also limits the applicability of IS2 for operational ice charting (see also the comment below).

We appreciate the reviewers comment on this. Cloud cover is a major factor impacting the IS2 observation, and has not been ignored, but commented several times (L. 180, 186, 216, 225, 238), but as has also been highlighted by Reviewer#1, it should be discussed in further detail. We aim to include more on this in the revised paper, including a thorough description of when data was available/disturbed by clouds, and we will provide quantification of the percentage of photons removed in the pre-processing step.

We have discussed the impact of cloud cover on IS2 (and for operational ice charting) in more detail in Sect. 3.3.1.

Line 228: I don't agree that "this study shows the potential". The study only compares IS2 and manual ice charts on a couple of cases with ~50% accuracy. How can actually IS2 measurements be used in ice charting? Which weight should an ice analyst give them compared to icebreaker observations? How accessible IS2 data would be due to cloud cover and latency? These and other questions need to be answered to show the potential.

We appreciate the reviewer's comment on this, and while we hoped we had answered some of these questions already (latency, cloud cover etc.), we see the need for even more discussion on these aspects. We aim to include this in the revised paper. The goal of our study is to demonstrate the utility of IS2 data for the community of end users operating in ice-covered water, and we hope that our paper will provide motivation for new IS2 elevation products with lower latencies in the future.

More on this has been included in Sect. 3.4 (Future outlook).

Line 233: "to estimate the ice conditions in certain areas for planning purpose"

Thank you. Corrected.

Line 255: Can you specify how IS2 could be used?

An example for ice charting in general has already been provided in L. 234-235: "An example of such planning would be compiling the statistical information on ice conditions required by the International Maritime Organisation (IMO) polar code to create a regional climatology (IMO, 2020)."

We shall give more concrete examples in the revised manuscript. We have included additional examples/suggestions in the 'Future outlook' section (Sect. 3.4).

Line 262: Given low correlation between IS2 and DIR I would recommend to rephrase: "… we have showed that there is some level of correspondence between FIS DIR and height anomalies using geolocated …"

Noted, thank you. Added.

Lines 270 – 273: Please rephrase "…we find that in some cases along-track densities of relative …" and split into simpler sentences.

Noted, thank you. This has now been rephrased and reworked into a simpler sentence.

---

## Referee Report (RR1)

**General comments**

The manuscript has been greatly improved. Comparison with SAR imagery increases confidence in the value of IS2 measurements. Analysis of uncertainties and error sources is detailed. Nevertheless, feasibility of using FIS DIR for training the classification algorithm is still questionable.

I'll try to explain my concern from a different angle. In Section 2.4 you mention that clear separation of elevation anomalies (EA) into three groups appears only when you use P90 and you even use only 5% of anomalies with the highest values for computing the thresholds. But it means that only 5% of the classified IS2 DIR correctly correspond to FIS DIR. The rest, 95% of EA cannot be classified in accordance FIS DIR by definition.

To illustrate I use your plot with histograms of EA separated by DIR level. I draw a sum of histograms as a thick black line – that's a distribution of *a priori* unknown EA. It is not accurate as I draw it by hands, but it gives a realistic representation. I add dashed lines at location of your thresholds. I mark with green, blue, red patches the sections of EA histograms that are correctly classified. If you compare the area of these patches with the rest of the histogram, you can also see that it constitutes approximately 5%. The rest of EA (area under the thick black curve) are classified incorrectly or not classified.

[Figure]

The fact that less than 5% IS2 DIR correspond to FIS DIR must be clearly presented in the manuscript.

In my opinion we can confidently conclude that the available FIS DIR cannot be used for training or testing of DIR classification from IS2 data. Therefore, either the title should be changed as proposed earlier to "Comparison of IS2 elevation anomalies with FIS DIR…", or a clear statement that more data (e.g. more FIS DIR observations or quantitative SAR data) is needed for developing a reliable DIR level classification algorithm should be added to conclusions.

**Detailed comments**

**Line 180.** Higher reliability of the classification is not proven. Only the classification of the 5% of points is more reliable. Classification of the rest of the data (extrapolation ability) may become less reliable when you decrease the amount of data. Either remove "reliable" or prove that it is more reliable by splitting the training dataset in two random parts, training the classifier on one part of data and applying it to another part of data.

**Figure 6.**
The intention behind the figure is very good but it needs to be improved.
SAR images are too small for visual analysis, profile plots are to busy to distinguish anything.
Please enlarge SAR images and reduce size of profiles and remove excessive data from profiles (e.g. present only one beam on a plot) and decrease the range of Y-axis down to [-0.5, 1] in order to stretch the profiles vertically.
Titles on planes f and h should be corrected to 27 March 2019.

**Line 470.** The argument that "IS2 seems to carry similar information as the S1 images, and the FIS DIR are based partly on icebreaker observations and partly on SAR imagery" is obviously weak and cannot be used to justify usage of FIS DIRs to train or validate algorithms. As mentioned above, only 5% of the classified anomalies correspond to FIS DIR, making it unusable.

---

## Referee Report (RR2)

**Review of Revisions to: Estimation of degree of sea ice ridging in the Bay of Bothnia based on geolocated photon heights from ICESat-2**

**1   General Comments**

**1.1   Synopsis of Changes**

The authors have introduced some more quantitative aspects to their analysis and have evaluated against SAR data, which looks to me to be well done but this should be corroborated by Reviewer 2. New sections have been introduced on ridge anisotropy and cloud-cover in response to my previous comments, which I find to be satisfactory. However several small issues have been introduced within the figures. As with my first review, I recommend this manuscript for publication pending the minor modifications suggested below.

**2   Specific Comments**

**2.1   Figures**

Fig. 3: PDFs have units that are the inverse of the x axis, such that the area under the curves equal to one and are dimensionless. So the y axis shouldn't be % if the x axis is in m.

Figure 5: Colorbar ticks should be coincident with color transitions - this issue was present in the originally submitted manuscript but I didn't pick it up. This should be fixed prior to publication.

Figure 10: The lon/lat ticks here are so small as to be almost unreadable. Please enlarge.

Figure 4: Not a problem that 'up' on your maps aren't orientated North, but it is fairly standard to add a small annotation in the corner to indicate to the reader the North direction (when it is not upward). This is particularly important when your grid lines are feint and other maps (e.g. Fig 10) are North-orientated. Apologies again for not picking this up on the first review.

Figures 2, 6, 7, 8: Much has been written about rainbow color maps (such as `gist-rainbow` used here) in the earth sciences, and they should be avoided (e.g. Borland and Taylor, 2007; Crameri et al., 2020). They have the effect of implying sharp transitions in the data where they do not exist, and are difficult to read for those who are colorblind (0.5% of women and 8% of men worldwide). Many good alternatives are available, see Light and Bartlein (2004); Stauffer et al. (2015); Thyng et al. (2016)

**2.2   Text**

L137: You need units on your plus/minus figure (metres).

L211: Does the thicker snow on fast-ice smooth the surface more? It's not obvious to me that snow bedforms such as dunes and sastrugi would produce a smoother surface than a relatively level, bare FYI, particularly at the radar-wavelength scale. I think you should cite this or instead of 'This is expected as' go for 'we attribute this to...'.

L508: It's true that estimating ice type (FYI/MYI) is tricky, I think you're talking more generally about ice

properties. Perhaps change type -¿ properties.

L509: Change cm to centimeter

L510: It's a nice point that cm-scale roughness can confound radar estimates of large-scale roughness, but it should be cited.

L498: Should be near-real-time

**References**

Borland, D. and Taylor, R. M.: Rainbow color map (still) considered harmful, IEEE Computer Graphics and Applications, 27, 14–17, https://doi.org/10.1109/MCG.2007.323435, 2007.

Crameri, F., Shephard, G. E., and Heron, P. J.: The misuse of colour in science communication, Nature Communications, 11, 1–10, https://doi.org/10.1038/s41467-020-19160-7, URL `https://doi.org/10.1038/s41467-020-19160-7`, 2020.

Light, A. and Bartlein, P. J.: The end of the rainbow? color schemes for improved data graphics, Eos, 85, https://doi.org/10.1029/2004EO400002, 2004.

Stauffer, R., Mayr, G. J., Dabernig, M., and Zeileis, A.: Somewhere over the rainbow: How to make effective use of colors in meteorological visualizations, Bulletin of the American Meteorological Society, 96, 203–216, https://doi.org/10.1175/BAMS-D-13-00155.1, 2015.

Thyng, K. M., Greene, C. A., Hetland, R. D., Zimmerle, H. M., and DiMarco, S. F.: True colors of oceanography: Guidelines for effective and accurate colormap selection, https://doi.org/10.5670/oceanog.2016.66, 2016.

---

## Author Response (AR2)

**Our comments**

Editor:

Dear Dr. Hansen and co-authors,

I have received two reviews based on your revised version of the manuscript "Estimation of degree of sea ice ridging in the Bay of Bothnia based on geolocated photon heights from ICESat-2." Both reviewers agree that manuscript has been substantially improved but still recommend a few more minor revisions. Reviewer #1 has several specific comments outlined in their referee report. Reviewer #2 still has a concern about the FIS DIR validation dataset and has provided several suggestions to the authors for alleviating this concern.

Please submit a revised version of your manuscript addressing the specific concerns of Reviewer 1 and validation concern raised by Reviewer 2.

Thank you for submitting your work to The Cryosphere.

Steve

**We wish to thank both of the reviewers for their sharp eyes and invaluable comments, which have improved the manuscript. We have taken their comments into consideration, and hope that the Editor finds our revisions satisfactory.**

**In particular, we have:**
- **Updated the figures to alleviate the concerns of both Reviewer#1 and #2, which includes updated figures on the distributions (correcting the units), enlarging the SAR intercomparison figures, and improving the photon profiles as proposed by Reviewer report#1. Furthermore, we have amended the color scales according to the comments of Reviewer report#2.**
- **Corrected the text where new edits are proposed by the reviewers.**
- **Small corrections in the text for clarification or consistency (e.g. writing data as plural instead of singular form, as given by the guidelines).**

**Sincerely, on behalf on the co-authors,**
**Renée Mie Fredensborg Hansen**

Referee report #1 (Reviewer #2):

General comments
The manuscript has been greatly improved. Comparison with SAR imagery increases confidence in the value of IS2 measurements. Analysis of uncertainties and error sources is detailed. Nevertheless, feasibility of using FIS DIR for training the classification algorithm is still questionable.

**We thank the reviewer for the positive feedback, and for the detailed review, which have greatly improved the manuscript.**

I'll try to explain my concern from a different angle. In Section 2.4 you mention that clear separation of elevation anomalies (EA) into three groups appears only when you use P90 and you even use only 5% of anomalies with the highest values for computing the thresholds. But it means that only 5% of the classified IS2 DIR correctly correspond to FIS DIR. The rest, 95% of EA cannot be classified in accordance with FIS DIR by definition.

To illustrate I use your plot with histograms of EA separated by DIR level. I draw a sum of histograms as a thick black line – that's a distribution of a priori unknown EA. It is not accurate as I draw it by hands, but it gives a realistic representation. I add dashed lines at location of your thresholds. I mark with green, blue, red patches the sections of EA histograms that are correctly classified. If you compare the area of these patches with the rest of the histogram, you can also see that it constitutes approximately 5%. The rest of EA (area under the thick black curve) are classified incorrectly or not classified.

The fact that less than 5% IS2 DIR correspond to FIS DIR must be clearly presented in the manuscript.

**We thank the reviewer for this in-depth description of their concern. Indeed, it means that less than 5% IS2 DIR corresponds to FIS DIR. We must also point out that it is not expected that the entire distribution would overall be very different, since ridges (elevation anomalies) are expected to be in the higher end of the distribution, and the distribution also includes elevation anomalies covering level ice (which will be the more dominant ice type). We will clearly state this in the manuscript for clarification.**

**We have included the following sentence in the end of Section 2.4:**
**"We note, that the intervals are based on the 5% highest elevation anomalies (using 95th percentile data), as we assume the highest elevation anomalies will include information on the ridges. Were one to use all of the elevation anomalies, it would also include elevation anomalies from the level ice (as seen by the overlapping distributions in Fig.3a)."**

In my opinion we can confidently conclude that the available FIS DIR cannot be used for training or testing of DIR classification from IS2 data. Therefore, either the title should be changed as proposed earlier to "Comparison of IS2 elevation anomalies with FIS DIR…", or a clear statement that more data (e.g. more FIS DIR observations or quantitative SAR data) is needed for developing a reliable DIR level classification algorithm should be added to conclusions.

**We are happy to include a clear statement in the conclusions that this is required. We have included the following sentence in the conclusions:**

**"We note, that this methodology of estimating IS2 DIR is based on the 5% highest elevation anomalies, and that while the IS2 DIR follows the deformation information seen in the SAR frames (qualitative assessment), the IS2 DIR only follows FIS DIR for some cases. To develop a reliable DIR level classification algorithm from IS2, more FIS DIR observations or quantitative SAR data should be included. "**

Detailed comments

Line 180. Higher reliability of the classification is not proven. Only the classification of the 5% of points is more reliable. Classification of the rest of the data (extrapolation ability) may become less reliable when you decrease the amount of data. Either remove "reliable" or prove that it is more reliable by splitting the training dataset in two random parts, training the classifier on one part of data and applying it to another part of data.

**The following has been removed: "..., ".**

Figure 6.
The intention behind the figure is very good but it needs to be improved.
SAR images are too small for visual analysis, profile plots are too busy to distinguish anything. Please enlarge SAR images and reduce size of profiles and remove excessive data from profiles (e.g. present only one beam on a plot) and decrease the range of Y-axis down to [-0.5, 1] in order to stretch the profiles vertically. Titles on planes f and h should be corrected to 27 March 2019.

**We do not believe that only one photon/elevation anomaly profile of one beam will be sufficient, when you are looking at all three beams on the map, and since they are separated by ~3km, they are observing different surfaces. We have changed the set-up of the figures, to enlarge the SAR images (4 subplots in one, new Figure 6), and added a new figure (new Figure 7) on the photon/elevation anomaly profiles, with 3 subplots per photon profile (P1-P4) to include each individual subbeam. We have also combined the HH polarization figures (Figure 7 and 8) into one figure (new Figure 8). We shall shrink the Y-axis a little, but will keep it at [-0.5, 1.5] since we also aim to look at the photon profiles used to generate the elevation anomalies and they extend from -1.5 to 3 m (with 3 of the 4 photon profiles exceeding the Y-axis suggested by the reviewer).**

Line 470. The argument that "IS2 seems to carry similar information as the S1 images, and the FIS DIR are based partly on icebreaker observations and partly on SAR imagery" is obviously weak and cannot be used to justify usage of FIS DIRs to train or validate algorithms. As mentioned above, only 5% of the classified anomalies correspond to FIS DIR, making it unusable.

We thank the reviewer for this comment. It is however not uncommon to train algorithms with ice charts (e.g., Gegiuc et al. 2018), so usage of FIS DIR to train with has been done before -- albeit not on IS2. We have revised this sentence, to clarify that more data is required (or potentially use other data sources such as Sentinel-1), to allow for a more robust classification in the future.

The sentence now says:
"However, since the classification of this study is based only on the 5% highest elevation anomalies, either more FIS DIR data or quantitative S1 data should be included and used for training, to ensure a robust classification. "

Referee report #2 (Reviewer#1):

1 General Comments

1.1 Synopsis of Changes

The authors have introduced some more quantitative aspects to their analysis and have evaluated against SAR data, which looks to me to be well done but this should be corroborated by Reviewer #2. New sections have been introduced on ridge anisotropy and cloud-cover in response to my previous comments, which I find to be satisfactory. However several small issues have been introduced within the figures. As with my first review, I recommend this manuscript for publication pending the minor modifications suggested below.

**We thank the reviewer for the positive feedback. Indeed, some small issues has been introduced in the figures, hence based on these comments of this review (and the comments of referee report#1), we have amended the figures accordingly.**

2 Specific Comments

2.1 Figures

Fig. 3: PDFs have units that are the inverse of the x axis, such that the area under the curves equal to one and are dimensionless. So the y axis shouldn't be % if the x axis is in m.

**Indeed! We thank the reviewer for their sharp eye and for making us aware of this mistake. This has been corrected.**

Figure 5: Colorbar ticks should be coincident with color transitions - this issue was present in the originally submitted manuscript but I didn't pick it up. This should be xed prior to publication.

**Thank you, we have aligned the colorbar ticks with the color transitions. We have also amended Table 3 and the text to fit with the values now aligned with the color transitions.**

Figure 10: The lon/lat ticks here are so small as to be almost unreadable. Please enlarge.

**Thank you, we have now enlarged it.**

Figure 4: Not a problem that `up' on your maps aren't orientated North, but it is fairly standard to add a small annotation in the corner to indicate to the reader the North direction (when it is not upward). This is particularly important when your grid lines are faint and other maps (e.g. Fig 10) are North-orientated. Apologies again for not picking this up on the first review.

**We thank the reviewer for this comment. We have included a north arrow (either inside the map or just beside the maps. Furthermore, we have made sure that all the**

**maps with IS2/FIS and IS2/SAR frames use the same projection (Transverse Mercator) and that the latitude/longitude labels are larger now. The gridlines are still faint (since we do not want them to take up too much of the figure, especially over the SAR frames), but we have made the SAR figures larger, so that the cyan-colored gridlines should be easier to distinguish now as well.**

Figures 2, 6, 7, 8: Much has been written about rainbow color maps (such as gist-rainbow used here) in the earth sciences, and they should be avoided (e.g. Borland and Taylor, 2007; Crameri et al., 2020). They have the effect of implying sharp transitions in the data where they do not exist, and are difficult to read for those who are colorblind (0.5% of women and 8% of men worldwide). Many good alternatives are available, see Light and Bartlein (2004); Stauer et al. (2015); Thyng et al. (2016)

**Yes, indeed. We actually discussed this in the beginning of writing the manuscript. However, it is crucial that we use a color scale with as many color transitions as possible to easily discriminate between differences in the elevation anomalies. Which was the rationale for picking this colormap at first. We thank the reviewer for providing us with links to the various studies that have looked into such alternatives, and have decided to use 'magma' instead based on the studies the reviewer suggested. We hope this is satisfactory. Unfortunately, when using the sequential colormaps (and not miscellaneous), there are fewer options available with a large range of colours. While the 'magma' colorbar seems to have enough variety to show the differences of the elevation anomalies, it however also means that some of the areas with low elevation anomalies are almost black, which in turn makes it difficult to see them on the SAR images. We have highlighted areas where it is important to notice these low elevation anomalies (e.g., Figure 2, region 2A). However, the focus of our study is mostly on the higher elevation anomalies (the bright values) which are easy to distinguish using the 'magma' colormap.**

2.2 Text

L137: You need units on your plus/minus figure (metres).

**Thank you, we have now included this.**

L211: Does the thicker snow on fast-ice smooth the surface more? It's not obvious to me that snow bedforms such as dunes and sastrugi would produce a smoother surface than a relatively level, bare FYI, particularly at the radar-wavelength scale. I think you should cite this or instead of `This is expected as' go for `we attribute this to...'.

**We have rewritten the sentence to: "We attribute this to regions of fast ice primarily consisting of smooth level ice represented by small differences in the elevation anomalies, and the fact that level ice overall carries more snow than drift ice, which could smooth the surface even further."**

L508: It's true that estimating ice type (FYI/MYI) is tricky, I think you're talking more generally about ice properties. Perhaps change type -> properties.

**Thank you, this has been changed accordingly.**

L509: Change cm to centimeter

**Thank you, this has been corrected.**

L510: It's a nice point that cm-scale roughness can confound radar estimates of large-scale roughness, but it should be cited.

**The following references has been cited:**

**Manninen, A.: Multiscale Surface Roughness and Backscattering, Progress In Electromagnetics Research, 16, 175–203, https://doi.org/10.2528/PIER96060700, 1997.**

L498: Should be near-real-time

**Thank you, this has been corrected.**

References

Borland, D. and Taylor, R. M.: Rainbow color map (still) considered harmful, IEEE Computer Graphics and Applications, 27, 14{17, https://doi.org/10.1109/MCG.2007.323435, 2007.

Crameri, F., Shephard, G. E., and Heron, P. J.: The misuse of colour in science communication, Nature Communications, 11, 1{10, https://doi.org/10.1038/s41467-020-19160-7, https://doi.org/10.1038/s41467-020-19160-7, 2020.

Light, A. and Bartlein, P. J.: The end of the rainbow? color schemes for improved data graphics, Eos, 85, https://doi.org/10.1029/2004EO400002, 2004.

Stauer, R., Mayr, G. J., Dabernig, M., and Zeileis, A.: Somewhere over the rainbow: How to make effective use of colors in meteorological visualizations, Bulletin of the American Meteorological Society, 96, 203{216, https://doi.org/10.1175/BAMS-D-13-00155.1, 2015.

Thyng, K. M., Greene, C. A., Hetland, R. D., Zimmerle, H. M., and DiMarco, S. F.: True colors of oceanography: Guidelines for effective and accurate colormap selection, https://doi.org/10.5670/oceanog.2016.66,2016.

**References**

Gegiuc, A., Similä, M., Karvonen, J., Lensu, M., Mäkynen, M., and Vainio, J.: Estimation of degree of sea ice ridging based on dual-polarized C-band SAR data, The Cryosphere, 12, 343–364, https://doi.org/10.5194/tc-12-343-2018, 2018.